# On the Identifiability of Nonlinear ICA: Sparsity and Beyond

**Yujia Zheng**[1,2], **Ignavier Ng**[1], **Kun Zhang**[1,2]
[1] Carnegie Mellon University
[2] Mohamed bin Zayed University of Artificial Intelligence
{yujiazh, ignavierng, kunz1}@cmu.edu

## Abstract

Nonlinear independent component analysis (ICA) aims to recover the underlying independent latent sources from their observable nonlinear mixtures. How to make the nonlinear ICA model identifiable up to certain trivial indeterminacies is a long-standing problem in unsupervised learning. Recent breakthroughs reformulate the standard independence assumption of sources as conditional independence given some auxiliary variables (e.g., class labels and/or domain/time indexes) as weak supervision or inductive bias. However, nonlinear ICA with unconditional priors cannot benefit from such developments. We explore an alternative path and consider only assumptions on the mixing process, such as *Structural Sparsity*. We show that under specific instantiations of such constraints, the independent latent sources can be identified from their nonlinear mixtures up to a permutation and a component-wise transformation, thus achieving nontrivial identifiability of nonlinear ICA without auxiliary variables. We provide estimation methods and validate the theoretical results experimentally. The results on image data suggest that our conditions may hold in a number of practical data generating processes.

## 1 Introduction

Nonlinear independent component analysis (ICA) is fundamental in unsupervised learning. It generalizes linear ICA (Comon, 1994) to identify latent sources from observations, which are assumed to be a nonlinear mixture of the sources. For an observed vector $\mathbf{x}$, nonlinear ICA expresses it as $\mathbf{x} = \mathbf{f}(\mathbf{s})$, where $\mathbf{f}$ is an unknown invertible mixing function, and $\mathbf{s}$ is a latent random vector representing the (marginally) independent sources. The goal is to recover function $\mathbf{f}$ as well as sources $\mathbf{s}$ from the observed mixture $\mathbf{x}$ up to certain indeterminacies. While nonlinear ICA is of general interest in a variety of tasks, such as disentanglement (Lachapelle et al., 2022) and unsupervised learning (Oja, 2002), its identifiability has been a long-standing problem for decades. The key difficulty is that, without additional assumptions, there exist infinite ways to transform the observations into independent components while still mixed w.r.t. the sources (Hyvärinen and Pajunen, 1999).

To deal with this challenge, existing works introduce the auxiliary variable $\mathbf{u}$ (e.g., class label, domain index, time index) and assume that sources are conditionally independent given $\mathbf{u}$ (Hyvärinen and Morioka, 2016, 2017; Hyvärinen et al., 2019; Khemakhem et al., 2020; Lachapelle et al., 2022). Most of them require the auxiliary variable to be observable, while clustering-based methods (Willetts and Paige, 2021) and those for time series (Hälvä and Hyvärinen, 2020; Hälvä et al., 2021; Yao et al., 2021) are exceptions. These works impose mild restrictions on the mixing process but require many distinct values of $\mathbf{u}$ for identifiability, which might restrict their practical applicability. Motivated by this, Yang et al. (2022) reduce the number of required distinct values by strengthening the functional assumptions on the mixing process. As described, all these results are based on conditional independence of the sources given auxiliary variable $\mathbf{u}$, instead of the standard marginal

independence assumption, to achieve identifiability in specific tasks, thanks to the weak supervision provided by the auxiliary variables. The identifiability of the original setting without auxiliary variables, which is crucial for unsupervised learning, stays unsolved.

Instead of introducing auxiliary variables, there exists the possibility to rely on further restrictions on the mixing functions to achieve identifiability, but limited results are available in the literature over the past two decades. With the assumption of conformal transformation, Hyvärinen and Pajunen (1999) proved identifiability of the nonlinear ICA solution up to rotation indeterminacy for only two sources. Another result is shown for the mixture that consists of component-wise nonlinearities added to a linear mixture (Taleb and Jutten, 1999).

In this work, we aim to show the identifiability of nonlinear ICA with unconditional priors. We prove that without any kind of auxiliary variables, the latent sources can be identified from nonlinear mixtures up to a component-wise invertible transformation and a permutation under assumptions only on the mixing process. Specifically, we mainly focus on the *structural sparsity* condition w.r.t. the structure from sources to observed variables, i.e., the support of the Jacobian matrix of the mixing function. Under structural conditions on the mixing process, one could identify the true sources by sparsity regularization during estimation (Thm. 1). By removing rotation indeterminacy, structural sparsity benefits the identifiability of linear ICA with Gaussian sources as well, which was previously thought to be unsolvable (Prop. 1, Thm. 2). Besides, we also develop another identifiability theory based on *Independent Influences*, which is an analogous concept of "sparsity" regarding the individual influencing process instead of the support (Prop. 2). We establish, to the best of our knowledge, one of the first identifiability results of the original nonlinear ICA model in a fully unsupervised setting.

Moreover, we also provide identifiability results for the undercomplete case, where the number of observed variables is larger than that of sources (Thm. 2, Thm. 3). This is of great practical interest and cannot be handled with previous conditions on nonlinear ICA, even with the help of auxiliary variables (Hyvärinen et al., 2019). We validate our theoretical claims experimentally, and the results on the image dataset suggest that our conditions appear to be reasonable for practical data generating processes.

## 2 Preliminaries

We consider the following data-generating process of ICA

$$p_{\mathbf{s}}(\mathbf{s}) = \prod_{i=1}^{n} p_{\mathbf{s}_i}(\mathbf{s}_i), \tag{1}$$

$$\mathbf{x} = \mathbf{f}(\mathbf{s}), \tag{2}$$

where $n$ denotes the number of latent sources, $\mathbf{x} = (\mathbf{x}_1, \ldots, \mathbf{x}_n)$ denotes the observed random vector, $\mathbf{s} = (\mathbf{s}_1, \ldots, \mathbf{s}_n)$ is the latent random vector representing the marginally independent sources, $p_{\mathbf{s}_i}$ is the marginal probability density function (PDF) of the $i$-th source $s_i$, $p_{\mathbf{s}}$ is the joint PDF of random vector $\mathbf{s}$, and $\mathbf{f} : \mathbf{s} \to \mathbf{x}$ denotes a nonlinear mixing function. For linear ICA, Eq. (2) is restricted as

$$\mathbf{x} = \mathbf{A}\mathbf{s}, \tag{3}$$

where $\mathbf{A} := [\mathbf{A}_{:,1} \cdots \mathbf{A}_{:,n}]$ denotes the mixing matrix. The goal of ICA is to learn an estimated unmixing function $\hat{\mathbf{f}}^{-1} : \mathbf{x} \to \hat{\mathbf{s}}$ (or alternatively in the linear case, an estimated unmixing matrix $\hat{\mathbf{A}}^{-1}$), such that $\hat{\mathbf{s}} = (\hat{\mathbf{s}}_1, \ldots, \hat{\mathbf{s}}_n)$ consists of independent estimated sources.

**Notations.** Throughout this work, for any matrix $\mathbf{M}$, we use $\mathbf{M}_{i,:}$ to refer to its $i$th row, and $\mathbf{M}_{:,j}$ to indicate its $j$th column. For any set of indices $\mathcal{S} \subset \{1, \ldots, m\} \times \{1, \ldots, n\}$, we analogously denote $\mathcal{S}_{i,:} := \{j \mid (i,j) \in \mathcal{S}\}$ and $\mathcal{S}_{:,j} := \{i \mid (i,j) \in \mathcal{S}\}$. We denote by $|\cdot|$ the cardinality of a set. We also define the following technical notations.

**Definition 1.** *Given a subset $\mathcal{S} \subseteq \{1, \ldots, n\}$, the subspace $\mathbb{R}^n_{\mathcal{S}}$ is defined as*

$$\mathbb{R}^n_{\mathcal{S}} := \{z \in \mathbb{R}^n \mid i \notin \mathcal{S} \implies z_i = 0\}. \tag{4}$$

In other words, $\mathbb{R}^n_{\mathcal{S}}$ refers to the subspace of $\mathbb{R}^n$ indicated by an index set $\mathcal{S}$. In the following, we define the support of a matrix.

**Definition 2.** *The support of matrix $\mathbf{M} \in \mathbb{R}^{m \times n}$ is defined as*

$$\text{supp}(\mathbf{M}) := \{(i,j) \mid \mathbf{M}_{i,j} \neq 0\}. \tag{5}$$

With slight abuse of notation, we also reuse the notation supp to denote the support of a matrix-valued function, depending on the context.

**Definition 3.** *The support of function* $\mathbf{M} : \Theta \to \mathbb{R}^{m \times n}$ *is defined as*

$$\text{supp}(\mathbf{M}(\Theta)) := \{(i, j) \mid \exists \theta \in \Theta, \mathbf{M}(\theta)_{i,j} \neq 0\}. \tag{6}$$

## 3 Identifiability with Structural Sparsity

### 3.1 Nonlinear ICA with Unconditional Priors: Structural Sparsity

In this section, we consider the identifiability of nonlinear ICA with unconditional priors, where the mixing function $\mathbf{f}$ in Eq. (2) is nonlinear. Different from recent breakthroughs in identifiable nonlinear ICA with side information (Hyvärinen et al., 2019; Lachapelle et al., 2022), in our setting, sources $\mathbf{s}$ in Eq. (1) do not need to be mutually conditionally independent given an auxiliary variable. Instead, our setting is consistent with the original ICA problem based on a general marginal independence assumption of sources. Besides, no particular form of the source distribution is assumed, which is different from most works that directly assume an exponential family (Hyvärinen et al., 2019). The goal of identifiable nonlinear ICA is to estimate the unmixing function $\hat{\mathbf{f}}^{-1}$ so that the independent sources are identified up to a certain indeterminacy, which is usually a composition of a component-wise invertible transformation and a permutation (Hyvärinen and Pajunen, 1999). We formulate the structural sparsity condition below and present the identifiability result with its proof shown in Appx. A.1. It is worth noting that Lachapelle et al. (2022) leverage sparse mechanism between latents and auxiliary variables for disentanglement, by which part of the assumptions and proof technique are inspired. For brevity, we denote $\mathcal{F}$ and $\hat{\mathcal{F}}$ as the support of the Jacobian $\mathbf{J_f}(\mathbf{s})$ and $\mathbf{J_{\hat{f}}}(\hat{\mathbf{s}})$, respectively. And $\mathbf{T}$ is a matrix with the same support of $\mathbf{T}(\mathbf{s})$ in $\mathbf{J_{\hat{f}}}(\hat{\mathbf{s}}) = \mathbf{J_f}(\mathbf{s})\mathbf{T}(\mathbf{s})$.

**Theorem 1.** *Let the observed data be sampled from a nonlinear ICA model as defined in Eqs. (1) and (2). Suppose the following assumptions hold:*

  i. *Mixing function* $\mathbf{f}$ *is invertible and smooth. Its inverse is also smooth.*

  ii. *For all* $i \in \{1, \dots, n\}$ *and* $j \in \mathcal{F}_{i,:}$, *there exist* $\{\mathbf{s}^{(\ell)}\}_{\ell=1}^{|\mathcal{F}_{i,:}|}$ *and* $\mathbf{T}$ *s.t.* $\text{span}\{\mathbf{J_f}(\mathbf{s}^{(\ell)})_{i,:}\}_{\ell=1}^{|\mathcal{F}_{i,:}|} = \mathbb{R}_{\mathcal{F}_{i,:}}^n$ *and* $\left[\mathbf{J_f}(\mathbf{s}^{(\ell)})\mathbf{T}\right]_{j,:} \in \mathbb{R}_{\hat{\mathcal{F}}_{i,:}}^n$.

  iii. $|\hat{\mathcal{F}}| \leq |\mathcal{F}|$.

  iv. *(Structural Sparsity) For all* $k \in \{1, \dots, n\}$, *there exists* $\mathcal{C}_k$ *such that*

$$\bigcap_{i \in \mathcal{C}_k} \mathcal{F}_{i,:} = \{k\}.$$

*Then* $\mathbf{h} := \hat{\mathbf{f}}^{-1} \circ \mathbf{f}$ *is a composition of a component-wise invertible transformation and a permutation.*

Assumption *ii* in a generic sense rules out a set of specific parameters to avoid ill-posed conditions such as the Jacobian is partially constant and is almost always to be satisfied asymptotically. Assumption *iii* corresponds to incorporating a suitable sparsity regularization term into the estimating process with no restriction on the ground truth. It helps to find the estimated mixing function with the minimal $L_0$ norm among all functions that allow the model to perfectly fit the true distribution of sources. Thus, it indicates that we should estimate the mixing process by maximizing its sparsity. Our discussion will focus on the assumption of structural sparsity, which is the core of our theory.

We start with the intuition of the proposed direction on exploiting sparsity on the mixing process for identifiability. As discussed in previous works (Hyvärinen and Pajunen, 1999), there exist infinitely many ways to preserve the independence among variables after mixing them up, such as Darmois construction (Darmois, 1951) and measure-preserving automorphism (MPA) (Hyvärinen and Pajunen, 1999). Besides, sources with non-Gaussian priors could be transferred to marginally independent Gaussians by trivial point-wise transformations (Hyvärinen et al., 2019). Thus, the existing strategy of exploiting independence and non-Gaussianity for identification fails in the nonlinear case. However, if the true structure is sparse enough, any alternative solution that produces any indeterminacy beyond component-wise transformations and permutations (e.g., mixtures or rotations) may correspond to a

denser structure. In the light of that, instead of restricting the functional class (e.g., post nonlinear models (Taleb and Jutten, 1999) or conformal maps (Hyvärinen and Pajunen, 1999)) or introducing auxiliary variables for extra structural dependencies between latent sources (Hyvärinen et al., 2019; Lachapelle et al., 2022), we leverage the sparsity pattern, i.e., the support of the Jacobian of the mixing function, to identify the sources.

The structural sparsity assumption (Assumption *iv* in Thm. 1) implies that, for every latent source $\mathbf{s}_i$, there exists a set of observed variable(s) such that $\mathbf{s}_i$ is the only latent source that participates in the generation of all observed variable(s) in the set. Graphically, for every latent source $\mathbf{s}_i$, there exists a set of observed variable(s) such that the intersection of their/its parent(s) is $\mathbf{s}_i$ (e.g., for $\mathbf{s}_1$ in Fig. 1, there exist $\mathbf{x}_1$ and $\mathbf{x}_4$ such that the intersection of their parents is $\mathbf{s}_1$.). This encourages the connections to be sparse enough to make the disentanglement of each source based on structural conditions possible. Together with the sparsity regularization during estimation (Assumption *iii*) and other mild conditions, Assumption *iv* illustrates a graphical pattern of nonlinear ICA models that could be identified by exploiting structural sparsity. It indicates a structural criterion of the sparsity that is needed for identifiability. Beyond that, structural sparsity is also of practical interest for the interpretability of results (Zhang et al., 2009) and identification of other latent variable models (Rhodes and Lee, 2021). Meanwhile, various versions of Occam's razor (e.g., faithfulness (Spirtes et al., 2000), minimality principle (Zhang, 2013), and frugality (Forster et al., 2020)) are also fundamental to the identifiability of the underlying causal structure. Readers may refer to additional discussion in Sec. 6.

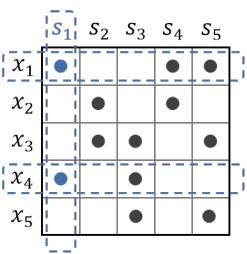

Figure 1: The structural sparsity assumption of Thm. 1, where the matrix represents $\mathrm{supp}(\mathbf{J_f}(\mathbf{s}))$.

With structural sparsity, Thm. 1 shows the identifiability of nonlinear ICA while maintaining the standard mutually marginal independence assumption of sources. This is consistent with the original setting of ICA and plays an important role in a more general range of unsupervised learning tasks, compared to previous works relying on conditional independence given auxiliary variables. Moreover, one may modify Thm. 1 to consider the generating process based on auxiliary variables by imposing similar restrictions on the mixing process. Different from previous works (Hyvärinen et al., 2019; Yang et al., 2022), this extension removes common restrictions on the required auxiliary variables, such as the number of distinct values (e.g., class labels), non-proportional variances, or sufficient variability. Therefore, although our main result (Thm. 1) focuses on nonlinear ICA with unconditional priors, it also provides potential insight into improving the flexibility of utilizing the auxiliary variable when it is available. As a trade-off, the additional assumptions on the sparsity might limit its usage.

Besides, under additional assumptions, we could reduce the indeterminacy in Thm. 1 and further identify sources up to a component-wise *linear* transformation. Most works in nonlinear ICA focus on the identifiability up to a component-wise invertible transformation and a permutation (Hyvärinen and Morioka, 2016; Khemakhem et al., 2020), which is also called *permutation-identifiability* in the literature of disentanglement (Lachapelle et al., 2022). This indeterminacy is trivial compared to the fundamental nonuniqueness of nonlinear ICA and is analogous to the indeterminacy involving rescaling and permutation in linear ICA (Hyvärinen and Pajunen, 1999). Recently, Yang et al. (2022) provided an identifiability result based on auxiliary variables that further reduces the indeterminacy of the component-wise nonlinear transformation to a linear one. Inspired by it but without auxiliary variables, we show conditions for the identifiability of nonlinear ICA with unconditional priors up to a component-wise linear transformation, with the proof in Appx. A.2. The reduced indeterminacy may give rise to a more informative disentanglement and open up the possibility for further improving the quality of recovery with only element-wise linear operations, such as the Hadamard product.

**Corollary 1.** *Let the observed data be sampled from a nonlinear ICA model as defined in Eqs. (1) and (2). Suppose the following assumptions hold:*

> *i. The function $\mathbf{h} := \hat{\mathbf{f}}^{-1} \circ \mathbf{f}$ is a composition of a component-wise invertible transformation and a permutation.*
>
> *ii. The mixing function $\mathbf{f}$ is volume-preserving.*
>
> *iii. The source distribution $p_{\mathbf{s}}(\mathbf{s})$ is a factorial multivariate Gaussian.*

*Then $\mathbf{h} := \hat{\mathbf{f}}^{-1} \circ \mathbf{f}$ is a composition of a component-wise linear transformation and a permutation.*

### 3.2 Removing Rotation Indeterminacy with Structural Sparsity

Rotation indeterminacy is one of the major obstacles to the identifiability of ICA. Linear ICA exploits the maximization of the non-Gaussianity of the estimated sources (e.g., Kurtois (Hyvärinen and Oja, 1997)). As a direct result, the typical assumption is that at most one of the sources can have Gaussian distribution. In contrast, for the nonlinear case, a trivial point-wise function could transform sources to have any marginal distribution including Gaussian thus invalidating the effect of non-Gaussianity. Therefore, removing rotation indeterminacy remains critical for the identifiability in the nonlinear case. For instance, "rotated-Gaussian" MPA produces nonuniqueness in nonlinear ICA due to rotation indeterminacy (Hyvärinen and Pajunen, 1999). It first maps the distribution to an isotropic Gaussian, then applies a rotation and maps it back without affecting its original distribution. To address it for identifiability, Yang et al. (2022) assume that there do not exist two conditionally independent Gaussians such that their variances are proportional.

In the previous section, we have introduced the structural sparsity for the identifiability of nonlinear ICA. One may wonder whether it straightforwardly leads to the identifiability of linear ICA with Gaussian sources (Gaussian ICA), which was previously perceived to be impossible due to its rotation indeterminacy. The idea is simple–any rotation of the true mixing matrix $\mathbf{A}$ will be less sparse if $\mathbf{A}$ satisfies the proposed structural sparsity assumption. While the joint distribution of Gaussian sources stays invariant across rotations, the sparsity of the mixing matrix (i.e., $L_0$ norm) keeps changing. We first consider Assumption *iv* in Thm. 1 for Gaussian ICA as an extension. Because the Jacobian of linear mixing function is fixed, Assumption *ii* of Thm. 1 does not directly hold in the linear Gaussian case. As a result, we cannot directly apply Thm. 1 here and therefore propose some alternative conditions:

**Proposition 1.** *Let the observed data be sampled from a linear ICA model defined in Eqs.* (1) *and* (3) *with Gaussian sources. Suppose the following assumptions hold:*

    *i. Mixing matrix $\mathbf{A}$ is invertible.*

    *ii. There exists a matrix $\hat{\mathbf{A}}$ s.t. for all $j \in \text{supp}(\mathbf{A})_{i,:}$, $\text{supp}(\hat{\mathbf{A}}\mathbf{A}^{-1})_{j,:} \in \mathbb{R}^n_{\text{supp}(\hat{\mathbf{A}})_{i,:}}$.*

    *iii. $|\text{supp}(\hat{\mathbf{A}})| \leq |\text{supp}(\mathbf{A})|$.*

    *iv. (Structural Sparsity) For all $k \in \{1, \ldots, n\}$, there exists $\mathcal{C}_k$ such that*

$$\bigcap_{i \in \mathcal{C}_k} \text{supp}(\mathbf{A}_{i,:}) = \{k\}.$$

*Then $\hat{\mathbf{A}} = \mathbf{A}\mathbf{D}\mathbf{P}$, where $\mathbf{D}$ is a diagonal matrix and $\mathbf{P}$ is a column permutation matrix.*

The proof is provided in Appx. A.3. Similar to Assumption *ii* of Thm. 1, Assumption *ii* of Prop. 1 excludes a specific set of parameters that makes structure-based identification ill-posed.

**The undercomplete case.** For the full identifiability of nonlinear ICA, it is noteworthy that the invertibility of the mixing function is technically essential in our result (Thm. 1). This invertibility implies that the number of observed variables is equal to that of sources. Although that setting is rather common in the literature, especially for the theory of nonlinear cases (Hyvärinen and Pajunen, 1999; Hyvärinen et al., 2019), undercomplete ICA (i.e., there are more observed variables), may be of more practical interest and more challenging (Joho et al., 2000). Therefore, we develop an alternative structural sparsity condition for the undercomplete case. As a trade-off, in the nonlinear case, the structural condition that removes the requirement of invertibility could only deal with the rotation indeterminacy. Thus we start from Gaussian ICA to introduce the condition.

**Definition 4.** *Given a matrix $\mathbf{S} \in \mathbb{R}^{m \times n}$, we define the function* overlap : $\mathbb{R}^{m \times n} \to \{0, 1\}^{m \times n}$ *as*

$$(\text{overlap}(\mathbf{S}))_{ij} = \begin{cases} 1 & \text{if } \mathbf{S}_{ij} = 1 \text{ and it is not the only nonzero entry in } \mathbf{S}_{i,:} \\ 0 & \text{otherwise.} \end{cases}$$

**Theorem 2.** *Let the observed data be sampled from a linear ICA model defined in Eqs.* (1) *and* (3) *with Gaussian sources. Differently, the number of observed variables (denoted as $m$) could be larger than that of the sources $n$, i.e., $m \geq n$. Suppose the following assumptions hold:*

    *i. The nonzero coefficients of the mixing matrix $\mathbf{A}$ are randomly drawn from a distribution that is absolutely continuous with respect to Lebesgue measure.*

ii. *The estimated mixing matrix $\hat{\mathbf{A}}$ has the minimal $L_0$ norm during estimation.*

iii. *(Structural Sparsity) Given $\mathcal{C} \subseteq \{1, 2, \ldots, n\}$ where $|\mathcal{C}| > 1$, let $\mathbf{A}_\mathcal{C} \in \mathbb{R}^{m \times |\mathcal{C}|}$ represents a submatrix of $\mathbf{A} \in \mathbb{R}^{m \times n}$ consisting of columns with indices $\mathcal{C}$. Then, for all $k \in \mathcal{C}$, we have*

$$\left| \bigcup_{k' \in \mathcal{C}} \mathrm{supp}(\mathbf{A}_{k'}) \right| - \mathrm{rank}(\mathrm{overlap}(\mathbf{A}_\mathcal{C})) > |\mathrm{supp}(\mathbf{A}_k)|.$$

*Then $\hat{\mathbf{A}} = \mathbf{A}\mathbf{D}\mathbf{P}$ with probability one, where $\mathbf{D}$ is a diagonal matrix and $\mathbf{P}$ is a column permutation matrix.*

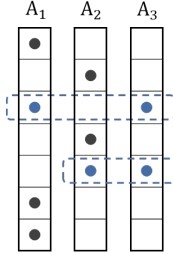

Figure 2: Assumption *iii* of Thm. 2.

The proof is provided in Appx. A.4. Intuitively, Assumption *iii* in Thm. 2 encourages the variability between the influences of each source. For each source, its influence on the observations should be as distinctive as possible, which is correlated with the sparsity of the mixing matrix. Fig. 2 is an illustration of the assumption. In that case, $k = 1$ and $\mathcal{C} = \{1, 2, 3\}$. Let $\mathbf{A}_\mathcal{C}$ denotes the submatrix shown in the figure, where $\left| \bigcup_{k' \in \mathcal{C}} \mathrm{supp}(\mathbf{A}_{k'}) \right| = 7$. Blue dotted square denotes $\mathrm{overlap}(\mathbf{A}_\mathcal{C})$ and we have $\mathrm{rank}(\mathrm{overlap}(\mathbf{A}_\mathcal{C})) = 2$. Thus, $\left| \bigcup_{k' \in \mathcal{C}} \mathrm{supp}(\mathbf{A}_{k'}) \right| - \mathrm{rank}(\mathrm{overlap}(\mathbf{A}_\mathcal{C})) = 5$ (the number of black dots), which is larger than $|\mathrm{supp}(\mathbf{A}_1)| = 4$. For the nonlinear case, we prove that it could help to distinguish spurious solutions due to the rotation indeterminacy, i.e., the "rotated-Gaussian" MPA (Defn. 2.5 in Gresele et al. (2021)). Gresele et al. (2021) prove that this class of MPAs could be ruled out with assumptions of conformal maps, non-Gaussianity and orthogonality of the Jacobian of the mixing function. Differently, we address it with the proposed structural sparsity condition (Assumption *iii* in Thm. 2). The corresponding theorem for the nonlinear case, with its proof given in Appx. A.5, is as follows:

**Theorem 3.** *Given a nonlinear ICA model defined in Eqs. (1) and (2), where $\mathbf{f}$ is the true mixing function. Consider $\hat{\mathbf{f}} = \mathbf{f} \circ \mathbf{G}^{-1} \circ \mathbf{U} \circ \mathbf{G}$, where $\mathbf{G}$ denotes an invertible Gaussianization[1] that maps the distribution to an standard isotropic (rotation-invariant) Gaussian, $\mathbf{U}$ denotes a rotation, and $\mathbf{G}^{-1}$ maps the distribution back to that before applying $\hat{\mathbf{U}} \circ \mathbf{G}$. If Assumptions i, ii and iii of Thm. 2 are satisfied by replacing $\mathbf{A}$ with $\mathbf{J}_{\mathbf{f}}(\mathbf{s})$ and $\hat{\mathbf{A}}$ with $\mathbf{J}_{\hat{\mathbf{f}}}(\mathbf{s})$, then function $\mathbf{h} := \hat{\mathbf{f}}^{-1} \circ \mathbf{f}$ is a composition of a component-wise invertible transformation and a permutation with probability one.*

Besides removing rotation indeterminacy, Thm. 3 also provides an extra insight for the full identifiability of nonlinear ICA, because all previous works assume that the nonlinear mixing function must be invertible (Hyvärinen and Morioka, 2016; Yang et al., 2022) even with the help of auxiliary variables.

## 4 Identifiability with Independent Influences

In the previous section, we have presented theoretical results based on conditions of structural sparsity. Alternatively, we show in this section that sparsity conditions on the support might not be necessary in some specific scenarios, and that one can leverage the independence (in a non-statistical sense) among the influences from different sources to the observations to achieve identifiability of nonlinear ICA. This implies sparse interactions between the individual influencing processes and could be viewed as an analogous concept of "sparsity" w.r.t. influences. We show that without any kind of auxiliary variables, the full identifiability of nonlinear ICA can also be provided given the condition of independent influences, thus complementing the results shown above from a different view of "sparsity".

**Nonlinear ICA with unconditional priors: Independent Influences.** We now specify the setting of the required estimating process. We sample the estimated sources $\hat{\mathbf{s}}$ from a multivariate Gaussian:

$$p_{\hat{\mathbf{s}}}(\hat{\mathbf{s}}) = \prod_{i=1}^{n} \frac{1}{Z_i} \exp\left( -\theta_{i,1} \hat{s}_i - \theta_{i,2} \hat{s}_i^2 \right), \tag{7}$$

---

[1]One example is described in (Gresele et al., 2021), i.e., a composition of the element-wise CDFs of a smooth factorised density and a Gaussian, respectively.

where $Z_i > 0$ is a constant. The sufficient statistics $\theta_{i,1} = -\frac{\mu_i}{\sigma_i^2}$ and $\theta_{i,2} = \frac{1}{2\sigma_i^2}$ are assumed to be linearly independent. We constraint the variances $\sigma_i^2$ to be distinct without loss of generality.

**Proposition 2.** *Let the observed data be sampled from a nonlinear ICA model as defined in Eqs.* (1), (2), *and* (7). *Suppose the following assumptions hold:*

i. *(Independent Influences): The influence of each source on the observed variables is independent of each other, i.e., each partial derivative $\partial \mathbf{f}/\partial s_i$ is independent of the other sources and their influences in a non-statistical sense.*

ii. *The mixing function $\mathbf{f}$ and its inverse are twice differentiable.*

iii. *The Jacobian determinant of mixing function can be factorized as $\det(\mathbf{J_f}(\mathbf{s})) = \prod_{i=1}^{n} y_i(s_i)$, where $y_i$ is a function that depends only on $s_i$. Note that volume-preserving transformation is a special case when $y_i(s_i) = 1, i = 1, \ldots, n$.*

iv. *During estimation, the columns of the Jacobian of the estimated unmixing function are regularized to be mutually orthogonal and with equal euclidean norm.*

*Then $\mathbf{h} := \hat{\mathbf{f}}^{-1} \circ \mathbf{f}$ is a composition of a component-wise invertible transformation and a permutation.*

The proof is included in Appx. A.6. Note that some part of the proof technique is inspired by or based on Lemma 1 in Yang et al. (2022). Assumption *iv* is a regularization during estimation and thus puts no restriction on the true generating process. We achieve it by optimizing an objective function with a designed regularization term, of which the property is as follows (proof in Appx. A.7):

**Proposition 3.** *The following inequality holds*

$$n \log \left( \frac{1}{n} \sum_{i=1}^{n} \left\| \frac{\partial \hat{\mathbf{f}}^{-1}}{\partial x_i} \right\|_2 \right) - \log \left| \det(\mathbf{J}_{\hat{\mathbf{f}}^{-1}}(\mathbf{x})) \right| \geq 0, \tag{8}$$

*with equality iff. $\mathbf{J}_{\hat{\mathbf{f}}^{-1}}(\mathbf{x}) = \mathbf{O}(\mathbf{x})\lambda(\mathbf{x})$, where $\mathbf{O}(\mathbf{x})$ is an orthogonal matrix and $\lambda(\mathbf{x})$ is a scalar.*

Regarding the assumption of independent influences, to build intuition, one can consider a filmmaking process in a safari park, of which the main characters are wild animals. The task of the boom operators here is to record the sound of animals. As there are many animals and microphones, the mixing (recording) process is highly dependent on the relative positions between them. Thus, the animal (source $s_i$), loosely speaking, influences the recording (mixing function $\mathbf{f}$) through its position. The moving direction and speed of them can be loosely interpreted as the partial derivative w.r.t. the $i$-th source, i.e. $\partial \mathbf{f}/\partial s_i$. Assuming that the wild animals are not cooperative enough to fine-tune their positions and speeds for a better recording and the safari park is not crowded, the influences of animals on the recording are generated independently by them in the sense that they are not affected by the others while moving. As a result of that generating process, the column vectors of the Jacobian of the mixing function are uncorrelated with each other and the partial derivative w.r.t. to each source is independent of other sources. Other practical scenarios include fields that adopt independent influences in process like orthogonal coordinate transformations (Gresele et al., 2021), such as dynamic control (Mistry et al., 2010) and structural geology (De Paor, 1983). It is worth noting that this assumption of independent influences is different from the orthogonality condition proposed in (Gresele et al., 2021). For instance, if the sample mean is not zero, uncorrelatedness differs from orthogonality. And we argue that uncorrelatedness as a specific way to achieve independence might be more sensible as a condition on the influences. Besides, in (Gresele et al., 2021), orthogonality, together with conformal map and the others, are proposed as conditions to rule out two specific types of spurious solutions (i.e., Darmois construction (Darmois, 1951) and "rotated-Gaussian" MPAs (Locatello et al., 2019)) without an identifiability result, although its empirical results are encouraging.

Having said that, a violation of the assumption of independent influences could be ascribed to a deliberate global adjustment of sources. For example, cinema audio systems are carefully adjusted to achieve a homogeneous sound effect on every audience. The position and orientation of each speaker are fine-tuned according to the others. In this case, it may be difficult to distinguish the influences from different speakers because of the fine-tuning. This may lead to a high degree of multicollinearity across the columns of Jacobian, thus violating the assumption of independent influences.

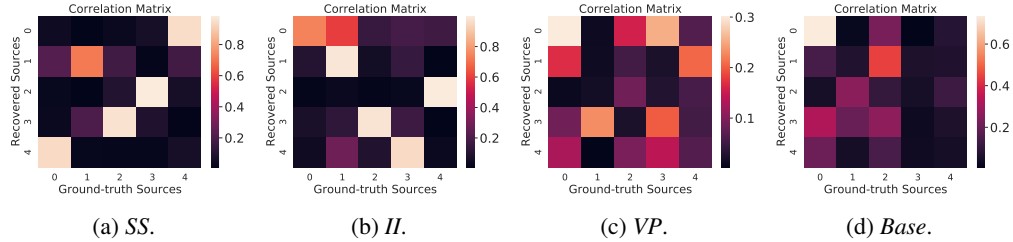

| (a) *SS.* | (b) *II.* | (c) *VP.* | (d) *Base.* |

Figure 3: Pearson correlation matrices between the ground-truth and the recovered sources.

Regarding Assumption *iii* in Prop. 2, we first note that mixing functions with factorial Jacobian determinants are of a much wider range as compared to component-wise transformations. To see this, a straightforward example is the volume-preserving transformation, which has been widely adopted in generative models (Zhang et al., 2021). In fact, all transformations with constant Jacobian determinant can be factorized w.r.t. the sources, which, generally speaking, are not component-wise. Besides, consistent with the empirical results in Yang et al. (2022), there exists other non-volume-preserving transformations with factorial Jacobian determinant [2]. Moreover, the volume-preserving assumption has been demonstrated to be helpful to weaken the requirement of auxiliary variables, specifically by reducing the number of required labels (Sorrenson et al., 2020; Yang et al., 2022). Prop. 2 takes this further by imposing a weaker version, i.e., Assumption (iii), which, together with the other constraints, removes the need for any auxiliary variable.

## 5   Experiments

To validate the proposed theory of the identifiability of nonlinear ICA with unconditional priors, we conduct experiments based on our main condition, i.e., structural sparsity (Thm. 1), as well as the condition of the independent influences (Prop. 2).

**Setup.**   For the required regularization during estimation, we consider a regularized maximum-likelihood approach with the following objective: $\mathcal{L}(\hat{\mathbf{f}}^{-1}; \mathbf{x}) = \mathbb{E}_{\mathbf{x}} \left[ \log p_{\hat{\mathbf{f}}^{-1}}(\mathbf{x}) - \lambda \mathbf{R}(\hat{\mathbf{f}}^{-1}, \mathbf{x}) \right]$, where $\lambda$ is a regularization parameter and $\mathbf{R}(\hat{\mathbf{f}}^{-1}, \mathbf{x})$ is the regularization term. For Thm. 1, we use $L_1$ norm as an approximation of the $L_0$ norm for efficiency (Donoho and Elad, 2003), therefore $\mathbf{R}(\hat{\mathbf{f}}^{-1}, \mathbf{x}) := \|\mathbf{J}_{\hat{\mathbf{f}}^{-1}}(\mathbf{x})\|_1$; for Prop. 2, $\mathbf{R}(\hat{\mathbf{f}}^{-1}, \mathbf{x})$ corresponds to LHS of Eq. (8). Following (Sorrenson et al., 2020; Yang et al., 2022), we train a GIN (Sorrenson et al., 2020) to maximize the the objective function $\mathcal{L}(\hat{\mathbf{f}}^{-1}; \mathbf{x})$.

**Ablation study.**   We conduct an ablation study to verify the necessity of the proposed assumptions. Specifically, we focus on the following models that correspond to different assumptions: *(SS)* The assumption of structural sparsity, as well as other assumptions in Thm. 1, are satisfied; *(II)* The assumption of independent influences, as well as other assumptions in Prop. 2, are satisfied; *(VP)* Compared to *II*, only the assumption of independent influences is violated while the other assumptions (e.g., factorizable Jacobian determinant) are still satisfied; *(Base)* The vanilla baseline. Compared to *VP*, the (un)mixing function is not restricted to having factorizable Jacobian determinants. The data are generated according to the required assumptions.

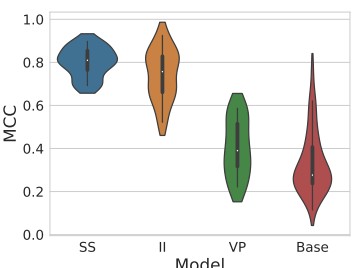

Figure 4: Ablation study.

We also conduct comparison between the assumption of independent influences and orthogonality in (Gresele et al., 2021), which are presented in Appx. B together with experimental settings. Results for each model are summarised in Fig. 4. For evaluation, we use the mean correlation coefficient (MCC) between the true sources and the estimated ones (Hyvärinen and Morioka, 2016). One could observe that when the proposed assumptions are fully satisfied (*SS* or *II*), our model achieves the highest MCC on average. This indicates that it is actually possible to identify sources from highly nonlinear mixtures up to trivial indeterminacies only based on restrictions on the mixing process. Moreover,

---

[2]A toy example: the Jacobian determinant of mixing function w.r.t. $\left( x_1 = \frac{a s_1 s_2}{a + b s_2}, x_2 = \frac{b s_1}{b + a s_2} \right)$ is $abs_1$, where $a, b \neq 0$ are some constants.

regarding Prop. 2, the importance of Assumption *iii* is supported by the higher performance of *VP* compared to that of *Base*. The visualization of the Pearson correlation matrices is shown in Fig. 3.

**Stability.** To study the stability of the performance of identification w.r.t. different datasets varying the number of sources $n$. We test the model *SS* (Thm. 1) with different $n$. Visually, we find that *SS* consistently outperforms *Base* (Fig. 5). Meanwhile, when the number of sources increases, one could observe that the MCC of *Base* decreases while that of *SS* stays stable. The stable empirical performance further validates our theoretical claims about the identifiability with structural sparsity.

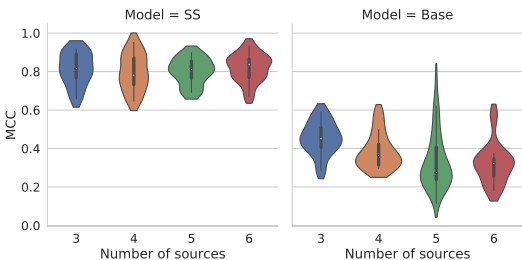

Figure 5: MCC w.r.t. different number of sources.

**Image dataset.** To study how reasonable the proposed theories are w.r.t. the practical generating process of observational data, we conduct experiments on the "Triangles" image dataset (Yang et al., 2022). The process of generating this dataset mimics the process of drawing triangles by humans: i) First, we sample the elements needed for humans to draw a monochrome triangle (i.e., rotation, width, height and grey level) from a factorial multivariate Gaussian distribution. Different from (Yang et al., 2022), we always sample from a single distribution in order to guarantee that all priors are unconditionally independent; ii) Then, for each pixel, we decide whether it locates inside the triangle based on the sampled

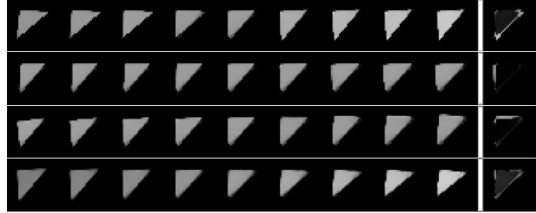

Figure 6: Identification results on Triangles. Each row represents a source identified by our model, with it varying from -2 to +2 SDs to illustrate its influence. The rightmost column is a heat map given by the absolute pixel difference between -1 and +1 SDs. Visually, the four rows correspond to rotation, width, height and gray level, respectively.

elements and assign its gray level accordingly. Therefore, the process is similar to human drawing triangles. Even though each image is generated from these semantic elements, the true generating process and sources are still unknown (e.g., a pixel could be (indirectly) influenced by multiple elements in a complicated way). We apply GIN with sparsity regularization as the estimating method. The visualization of the identified sources (Fig. 6) indicates that our conditions may hold in practice.

**IMA.** Recently, Gresele et al. (2021) assume orthogonality, conformal map, and the others in order to rule out Darmois construction and "rotated-Gaussian" MPAs. They formalize the orthogonality between columns of the Jacobian of the mixing function as independent mechanism analysis (*IMA*) and show empirically that it improves the identification of latent sources. To further explore their exciting results, we generate datasets according to the generating process and regularization term described in (Gresele et al., 2021). We sample the sources from the same distributions described in Appx. B and use GIN for training. From Fig. 7, one could observe that indeed *IMA* outperforms the baseline (*Base*) largely, which indicates that this condition is empirically helpful to the identifiability of nonlinear ICA. At the same time, MCC of *II* appears to be even higher than that of *IMA*, which might thanks to some additional constraints,

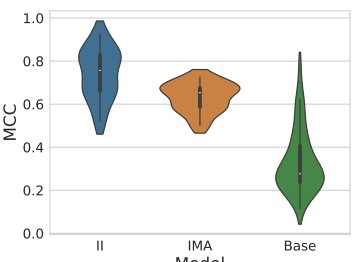

Figure 7: MCC w.r.t. independent influences (*II*), independent mechanism analysis (*IMA*), and the baseline model (*Base*).

such as the factorial volume. Meanwhile, it also supports the conjecture that *IMA* is a sensible condition for identification, and the identifiability based on it may be achieved with some additional assumptions, such as conformal maps (Gresele et al., 2021).

## 6  Discussion and Conclusion

**Sparsity Assumptions.** Sparsity assumptions have been widely used in various fields. For latent variable models, sparsity in the generating process plays an important role in the disentanglement or identification of latent factors both empirically and theoretically (Bing et al., 2020; Rohe and Zeng, 2020; Moran et al., 2021; Rhodes and Lee, 2021; Lachapelle et al., 2022). In causality, various versions of

Occam's razor have been proposed to serve as fundamental assumptions for identifying the underlying causal structure (Spirtes et al., 2000; Zhang, 2013; Raskutti and Uhler, 2018; Forster et al., 2020).

Formulated as a measure of the density of dependencies, sparsity assumptions are more likely to be held when the observations are actually influenced by the sources in a "simple" way. For example, in biology, analyses on ecological, gene-regulatory, metabolic, and other living systems find that active interactions may often be rather sparse (Busiello et al., 2017), even when these systems evolve with an unlimited number of complicated external stimuli. In physics, it is an important heuristic that a relatively small set of laws govern complicated observed phenomena. For instance, Einstein's theory of special relativity contains parsimonious relations between substances as an important heuristic to shave away the influence of ether compared to Lorentz's theory (Einstein, 1905; Nash, 1963).

However, sparsity is not an irrefutable principle and our assumption may fail in a number of situations. The most direct one could be a scenario with heavily entangled relations between sources and observations. Let us consider the example of animal filmmaking in Sec. 4, where people are recording the sound of animals in a safari park. If the filming location is restricted to a narrow area of the safari park and multiple microphones are gathered together, the recording of each microphone will likely be influenced by almost all animals. In such a case, the dependencies between the recording of microphones and the animals are rather dense and our sparsity assumption is most likely not valid.

At the same time, even when the principle of simplicity holds, our formulation of sparsity, which is based on the sparse interactions between sources and observations, may still fail. One reason for this is the disparity between mechanism simplicity and structural sparsity. To illustrate this, one could consider the effect of sunlight on the shadow angles at the same location. In this case, the sun's rays and the shadow angles act as the sources and the observations, respectively. Because rays of sunlight, loosely speaking, may be parallel to each other, the processes of them influencing the shadow angles may be almost identical. Thus, the influencing mechanism could be rather simple. On the other hand, each shadow angle is influenced by an unlimited number of the sun's rays, which indicates that the interactions between them may not be sparse, therefore violating our assumption. This sheds light on one of the limitations of our sparsity assumption, because the principle of simplicity could be formulated in several ways. Besides, these different formulations also suggest various possibilities for identifiability based on simplicity assumptions. Another proposed assumption, i.e., independent influences, may be one of the alternative formulations, and more works remain to be explored in the future.

**Conclusion.** We provide identifiability results for nonlinear ICA with unconditional priors, which serve as one of the first steps to solve a long-standing problem in unsupervised learning. In particular, we prove that the i.i.d. latent sources can be recovered up to a component-wise invertible transformation and a permutation with only conditions on the nonlinear mixing process (e.g., structural sparsity). Therefore it stays closer to the original notion of ICA that is based on the marginal independence assumption of latent sources, while previous works rely on conditional independence on auxiliary variables as weak supervision or inductive bias. Besides, by removing rotation indeterminacy, structural sparsity benefits the identifiability of Gaussian ICA as well, which was also thought to be unsolvable before. Moreover, the results on the undercomplete case are of great practical interest and introduce insight for extending identifiable nonlinear ICA to general real-world settings.

Our results on images illustrate the validity of the proposed conditions in practical data generating processes. In spite of this, it is possible that part of them is violated in several specific scenarios as discussed before. For example, the structural sparsity conditions do not apply to fully-connected structures, though the practical significance of such cases may be compromised by the lack of interpretability. As a complementary solution, we formulate the independent influences condition, which does not rely on the sparse structure of supports. While arguably natural in general, it could still be violated due to a deliberate global adjustment of sources for homogeneity. We argue that this is inevitably a trade-off between introducing auxiliary variables and imposing restrictions on the mixing process to achieve the identifiability, whose practical use depends on the scenario and information available. Future work includes further generalizing and validating our theory.

# Acknowledgements

We thank Lingjing Kong, Sébastien Lachapelle, and the anonymous reviewers for their constructive comments. This work was partially supported by the National Institutes of Health (NIH) under Contract R01HL159805, by the NSF-Convergence Accelerator Track-D award #2134901, by a grant from Apple Inc., and by a grant from KDDI Research Inc..

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
