# A Proofs

## A.1 Proof of Theorem 1

**Theorem 1.** *Let the observed data be sampled from a nonlinear ICA model as defined in Eqs. (1) and (2). Suppose the following assumptions hold:*

    *i. Mixing function $\mathbf{f}$ is invertible and smooth. Its inverse is also smooth.*

    *ii. For all $i \in \{1, \ldots, n\}$ and $j \in \mathcal{F}_{i,:}$, there exist $\{\mathbf{s}^{(\ell)}\}_{\ell=1}^{|\mathcal{F}_{i,:}|}$ and $\mathrm{T}$ s.t. $\mathrm{span}\{\mathbf{J_f}(\mathbf{s}^{(\ell)})_{i,:}\}_{\ell=1}^{|\mathcal{F}_{i,:}|} = \mathbb{R}^n_{\mathcal{F}_{i,:}}$ and $\left[\mathbf{J_f}(\mathbf{s}^{(\ell)})\mathrm{T}\right]_{j,:} \in \mathbb{R}^n_{\hat{\mathcal{F}}_{i,:}}$.*

    *iii. $|\hat{\mathcal{F}}| \leq |\mathcal{F}|$.*

    *iv. (Structural Sparsity) For all $k \in \{1, \ldots, n\}$, there exists $\mathcal{C}_k$ such that*

$$\bigcap_{i \in \mathcal{C}_k} \mathcal{F}_{i,:} = \{k\}.$$

*Then $\mathbf{h} := \hat{\mathbf{f}}^{-1} \circ \mathbf{f}$ is a composition of a component-wise invertible transformation and a permutation.*

*Proof.* Our goal here is to show that function $\mathbf{h} := \hat{\mathbf{f}}^{-1} \circ \mathbf{f}$ is a permutation with component-wise invertible transformation of sources, i.e., $\hat{\mathbf{f}} = \mathbf{f} \circ \mathbf{h}^{-1}(\mathbf{s})$. Let $\mathbf{D}(\mathbf{s})$ represents a diagonal matrix and $\mathbf{P}$ represent a permutation matrix. By using chain rule repeatedly, we write $\hat{\mathbf{f}} = \mathbf{f} \circ \mathbf{h}^{-1}(\mathbf{s})$ equivalently as

$$
\begin{aligned}
\mathbf{J}_{\hat{\mathbf{f}}}(\hat{\mathbf{s}}) &= \mathbf{J}_{\mathbf{f} \circ \mathbf{h}^{-1}}(\mathbf{h}(\mathbf{s})) \\
&= \mathbf{J}_{\mathbf{f} \circ \mathbf{g}^{-1} \circ \mathbf{P}^{-1}}(\mathbf{P}\mathbf{g}(\mathbf{s})) \\
&= \mathbf{J}_{\mathbf{f} \circ \mathbf{g}^{-1}}\left(\mathbf{P}^{-1}\mathbf{P}\mathbf{g}(\mathbf{s})\right)\mathbf{J}_{\mathbf{P}^{-1}}(\mathbf{P}\mathbf{g}(\mathbf{s})) \\
&= \mathbf{J}_{\mathbf{f} \circ \mathbf{g}^{-1}}(\mathbf{g}(\mathbf{s}))\mathbf{J}_{\mathbf{P}^{-1}}(\mathbf{P}\mathbf{g}(\mathbf{s})) \\
&= \mathbf{J}_{\mathbf{f}}\left(\mathbf{g}^{-1}\mathbf{g}(\mathbf{s})\right)\mathbf{J}_{\mathbf{g}^{-1}}(\mathbf{g}(\mathbf{s}))\mathbf{J}_{\mathbf{P}^{-1}}(\mathbf{P}\mathbf{g}(\mathbf{s})) \\
&= \mathbf{J}_{\mathbf{f}}(\mathbf{s})\mathbf{D}(\mathbf{s})\mathbf{P},
\end{aligned}
\tag{9}
$$

where $\mathbf{g}$ is an invertible element-wise function. Thus our goal is equivalent to show that

$$\mathbf{J}_{\hat{\mathbf{f}}}(\hat{\mathbf{s}}) = \mathbf{J}_{\mathbf{f}}(\mathbf{s})\mathbf{D}(\mathbf{s})\mathbf{P}. \tag{10}$$

Because $\mathbf{J}_{\hat{\mathbf{f}}}(\hat{\mathbf{s}})$ and $\mathbf{J}_{\mathbf{f}}(\mathbf{s})$ are both invertible, we have the following equation

$$\mathbf{J}_{\hat{\mathbf{f}}}(\hat{\mathbf{s}}) = \mathbf{J}_{\mathbf{f}}(\mathbf{s})\mathbf{T}(\mathbf{s}), \tag{11}$$

where $\mathbf{T}(\mathbf{s})$ is an invertible matrix.

Note that we denote $\mathcal{F}$ as the support of $\mathbf{J}_{\mathbf{f}}(\mathbf{s})$, $\hat{\mathcal{F}}$ as the support of $\mathbf{J}_{\hat{\mathbf{f}}}(\hat{\mathbf{s}})$ and $\mathcal{T}$ as the support of $\mathbf{T}(\mathbf{s})$. Besides, we denote $\mathrm{T}$ as a matrix with the support $\mathcal{T}$. According to Assumption ii, we have

$$\mathrm{span}\{\mathbf{J}_{\mathbf{f}}(\mathbf{s}^{(\ell)})_{i,:}\}_{\ell=1}^{|\mathcal{F}_{i,:}|} = \mathbb{R}^n_{\mathcal{F}_{i,:}}. \tag{12}$$

Since $\{\mathbf{J}_{\mathbf{f}}(\mathbf{s}^{(\ell)})_{i,:}\}_{\ell=1}^{|\mathcal{F}_{i,:}|}$ forms a basis of $\mathbb{R}^n_{\mathcal{F}_{i,:}}$, for any $j_0 \in \mathcal{F}_{i,:}$, we are able to rewrite the one-hot vector $e_{j_0} \in \mathbb{R}^n_{\mathcal{F}_{i,:}}$ as

$$e_{j_0} = \sum_{\ell \in \mathcal{F}_{i,:}} \alpha_\ell \mathbf{J}_{\mathbf{f}}(\mathbf{s}^{(\ell)})_{i,:}, \tag{13}$$

where $\alpha_\ell$ is the corresponding coefficient. Then

$$\mathrm{T}_{j_0,:} = e_{j_0}\mathrm{T} = \sum_{\ell \in \mathcal{F}_{i,:}} \alpha_\ell \mathbf{J}_{\mathbf{f}}(\mathbf{s}^{(\ell)})_{i,:}\mathrm{T} \in \mathbb{R}^n_{\hat{\mathcal{F}}_{i,:}}, \tag{14}$$

where the final "$\in$" follows from Assumption ii that each element in the summation belongs to $\mathbb{R}^n_{\hat{\mathcal{F}}_{i,:}}$. Thus

$$\forall j \in \mathcal{F}_{i,:}, \; \mathrm{T}_{j,:} \in \mathbb{R}^n_{\hat{\mathcal{F}}_{i,:}}. \tag{15}$$

Then we are able to draw connections between these supports according to Defn. 3

$$\forall (i,j) \in \mathcal{F}, \{i\} \times \mathcal{T}_{j,:} \subset \hat{\mathcal{F}}.\tag{16}$$

According to Assumption i, $\mathbf{T(s)}$ is an invertible matrix, indicating that it has a non-zero determinant. Representing the determinant of the matrix $\mathbf{T(s)}$ as its Leibniz formula yields

$$\det(\mathbf{T(s)}) = \sum_{\sigma \in \mathcal{S}_n} \left( \operatorname{sgn}(\sigma) \prod_{i=1}^{n} \mathbf{T(s)}_{i,\sigma(i)} \right) \neq 0,\tag{17}$$

where $\mathcal{S}_n$ is the set of $n$-permutations. Then there exists at least one term of the sum that is non-zero, i.e.,

$$\exists \sigma \in \mathcal{S}_n, \forall i \in \{1, \ldots, n\}, \operatorname{sgn}(\sigma) \prod_{i=1}^{n} \mathbf{T(s)}_{i,\sigma(i)} \neq 0.\tag{18}$$

This is equivalent to

$$\exists \sigma \in \mathcal{S}_n, \forall i \in \{1, \ldots, n\}, \mathbf{T(s)}_{i,\sigma(i)} \neq 0.\tag{19}$$

Then we can see that this $\sigma$ is in the support of $\mathbf{T(s)}$, which implies that

$$\forall j \in \{1, \ldots, n\}, \sigma(j) \in \mathcal{T}_{j,:}.\tag{20}$$

Together with Eq. (16), it follows that

$$\forall (i,j) \in \mathcal{F}, (i, \sigma(j)) \in \{i\} \times \mathcal{T}_{j,:} \subset \hat{\mathcal{F}}.\tag{21}$$

Denote

$$\sigma(\mathcal{F}) = \{(i, \sigma(j)) \mid (i,j) \in \mathcal{F}\}.\tag{22}$$

Then we have

$$\sigma(\mathcal{F}) \subset \hat{\mathcal{F}}.\tag{23}$$

According to Assumption iii, we can see that

$$|\hat{\mathcal{F}}| \leq |\mathcal{F}| = |\sigma(\mathcal{F})|.\tag{24}$$

Together with Eq. (23), we have

$$\sigma(\mathcal{F}) = \hat{\mathcal{F}}.\tag{25}$$

Suppose $\mathbf{T(s)} \neq \mathbf{D(s)P}$, then

$$\exists j_1 \neq j_2, \mathcal{T}_{j_1,:} \cap \mathcal{T}_{j_2,:} \neq \emptyset.\tag{26}$$

Besides, consider $j_3 \in \{1, \ldots, n\}$ such that

$$\sigma(j_3) \in \mathcal{T}_{j_1,:} \cap \mathcal{T}_{j_2,:}.\tag{27}$$

Because $j_1 \neq j_2$, we can assume $j_3 \neq j_1$ without loss of generality. A similar strategy has previously been used in (Lachapelle et al., 2022). By Assumption iv, there exists $\mathcal{C}_{j_1} \ni j_1$ such that $\bigcap_{i \in \mathcal{C}_{j_1}} \mathcal{F}_{i,:} = \{j_1\}$. Because

$$j_3 \notin \{j_1\} = \bigcap_{i \in \mathcal{C}_{j_1}} \mathcal{F}_{i,:},\tag{28}$$

there must exists $i_3 \in \mathcal{C}_{j_1}$ such that

$$j_3 \notin \mathcal{F}_{i_3,:}.\tag{29}$$

Because $j_1 \in \mathcal{F}_{i_3,:}$, we have $(i_3, j_1) \in \mathcal{F}$. Then according to Eq. (16), we have the following equation

$$\{i_3\} \times \mathcal{T}_{j_1,:} \subset \hat{\mathcal{F}}.\tag{30}$$

Notice that $\sigma(j_3) \in \mathcal{T}_{j_1,:} \cap \mathcal{T}_{j_2,:}$ implies

$$(i_3, \sigma(j_3)) \in \{i_3\} \times \mathcal{T}_{j_1,:}.\tag{31}$$

Then by Eqs. (31) and (30), we have

$$(i_3, \sigma(j_3)) \in \hat{\mathcal{F}},\tag{32}$$

which implies $(i_3, j_3) \in \mathcal{F}$ by Eq. (25) and Eq. (22), therefore contradicting Eq. (29). Thus, we prove by contradiction that $\mathbf{T(s)} = \mathbf{D(s)P}$. Replacing $\mathbf{T(s)}$ with $\mathbf{D(s)P}$ in Eq. (11), we prove Eq. (10), which is the goal. □

## A.2 Proof of Corollary 1

**Corollary 1.** *Let the observed data be sampled from a nonlinear ICA model as defined in Eqs. (1) and (2). Suppose the following assumptions hold:*

  *i. The function $\mathbf{h} := \hat{\mathbf{f}}^{-1} \circ \mathbf{f}$ is a composition of a component-wise invertible transformation and a permutation.*

  *ii. The mixing function $\mathbf{f}$ is volume-preserving.*

  *iii. The source distribution $p_\mathbf{s}(\mathbf{s})$ is a factorial multivariate Gaussian.*

*Then $\mathbf{h} := \hat{\mathbf{f}}^{-1} \circ \mathbf{f}$ is a composition of a component-wise linear transformation and a permutation.*

*Proof.* The proof technique of this corollary is inspired by Thm. 2 in Yang et al. (2022). According to Assumption iii, true sources $\mathbf{s}$ are from a factorial multivariate Gaussian distribution. Together with the estimated sources $\hat{\mathbf{s}}$ from the same type of distribution, we represent the densities of the true and estimated sources as

$$p_\mathbf{s}(\mathbf{s}) = \prod_{i=1}^{n} \frac{1}{Z_i} \exp\left(-\theta'_{i,1} s_i - \theta'_{i,2} s_i^2\right),$$
$$p_{\hat{\mathbf{s}}}(\hat{\mathbf{s}}) = \prod_{i=1}^{n} \frac{1}{Z_i} \exp\left(-\theta_{i,1} \hat{s}_i - \theta_{i,2} \hat{s}_i^2\right), \tag{33}$$

where $Z_i > 0$ is a constant. The sufficient statistics $\theta_{i,1}$ and $\theta_{i,2}$ are assumed to be linearly independent.

Applying the change of variable rule, we have $p_\mathbf{s}(\mathbf{s}) = p_{\hat{\mathbf{s}}}(\hat{\mathbf{s}})|\det(\mathbf{J_h}(\mathbf{s}))|$, which, by plugging in Eq. (33) and taking the logarithm on both sides, yields

$$\sum_{i=1}^{n} \log p_{s_i}(s_i) - \log|\det(\mathbf{J_h}(\mathbf{s}))| = -\sum_{i=1}^{n} \left(\theta_{i,1} h_i(\mathbf{s}) + \theta_{i,2} h_i(\mathbf{s})^2 + \log Z_i\right). \tag{34}$$

Because of Assumption ii and the corresponding estimating process, $\det(\mathbf{J_h}(\mathbf{s})) = 1$. Thus

$$\sum_{i=1}^{n} \log p_{s_i}(s_i) = -\sum_{i=1}^{n} \left(\theta_{i,1} h_i(\mathbf{s}) + \theta_{i,2} h_i(\mathbf{s})^2 + \log Z_i\right). \tag{35}$$

According to Assumption i, function $\mathbf{h}$ is a component-wise invertible transformation of sources, i.e.,

$$h_i(\mathbf{s}) = h_i(s_i). \tag{36}$$

Combining Eqs. (33) and (36) with Eq. (35), it follows that

$$\sum_{i=1}^{n} \left(-\theta'_{i,1} s_i - \theta'_{i,2} s_i^2\right) = -\sum_{i=1}^{n} \left(\theta_{i,1} h_i(s_i) + \theta_{i,2} h_i(s_i)^2 + \log Z_i\right).$$

Then we have

$$\theta'_{i,1} s_i + \theta'_{i,2} s_i^2 + \log Z_i = \theta_{i,1} h_i(s_i) + \theta_{i,2} h_i(s_i)^2.$$

Therefore, $h_i(s_i)$ is a linear function of $s_i$. $\square$

## A.3 Proof of Proposition 1

**Proposition 1.** *Let the observed data be sampled from a linear ICA model defined in Eqs. (1) and (3) with Gaussian sources. Suppose the following assumptions hold:*

  *i. Mixing matrix $\mathbf{A}$ is invertible.*

  *ii. There exists a matrix $\hat{\mathbf{A}}$ s.t. for all $j \in \text{supp}(\mathbf{A})_{i,:}$, $\text{supp}(\hat{\mathbf{A}}\mathbf{A}^{-1})_{j,:} \in \mathbb{R}^n_{\text{supp}(\hat{\mathbf{A}})_{i,:}}$.*

  *iii. $|\text{supp}(\hat{\mathbf{A}})| \leq |\text{supp}(\mathbf{A})|$.*

*iv.* *(Structural Sparsity)* *For all $k \in \{1, \ldots, n\}$, there exists $\mathcal{C}_k$ such that*

$$\bigcap_{i \in \mathcal{C}_k} \text{supp}(\mathbf{A}_{i,:}) = \{k\}.$$

*Then $\hat{\mathbf{A}} = \mathbf{A}\mathbf{D}\mathbf{P}$, where $\mathbf{D}$ is a diagonal matrix and $\mathbf{P}$ is a column permutation matrix.*

*Proof.* Let $\hat{\mathbf{A}} = \mathbf{A}\mathbf{T}$, where $\mathbf{T} \in \mathbb{R}^{n \times n}$ is an arbitrary matrix. Because our goal is to prove $\hat{\mathbf{A}} = \mathbf{A}\mathbf{D}\mathbf{P}$, it is equivalent to prove the following equation:

$$\mathbf{T} = \mathbf{D}\mathbf{P}. \tag{37}$$

For brevity of notation, we denote $\mathcal{A}$ as the support of $\hat{\mathbf{A}}$, $\hat{\mathcal{A}}$ as the support of $\mathbf{A}$ and $\mathcal{T}$ as the support of $\mathbf{T}$.

According to Assumption ii, we have

$$\forall j \in \text{supp}(\mathbf{A})_{i,:}, \; \text{supp}(\hat{\mathbf{A}}\mathbf{A}^{-1})_{j,:} \in \mathbb{R}^n_{\text{supp}(\hat{A})_{i,:}}, \tag{38}$$

which is equivalent to

$$\forall j \in \mathcal{A}_{i,:}, \; \mathbf{T}_{j,:} \in \mathbb{R}^n_{\hat{\mathcal{A}}_{i,:}}. \tag{39}$$

Then we have

$$\forall (i, j) \in \mathcal{A}, \; \{i\} \times \mathcal{T}_{j,:} \subset \hat{\mathcal{A}}. \tag{40}$$

According to Assumption i, $\mathbf{T}$ is an invertible matrix, indicating that it has a non-zero determinant. Representing the determinant of the matrix $\mathbf{T}$ as its Leibniz formula yields

$$\det(\mathbf{T}) = \sum_{\sigma \in \mathcal{S}_n} \left( \text{sgn}(\sigma) \prod_{i=1}^n \mathbf{T}_{i,\sigma(i)} \right) \neq 0, \tag{41}$$

where $\mathcal{S}_n$ is the set of $n$-permutations. Then there exists at least one term of the sum that is non-zero, i.e.,

$$\exists \sigma \in \mathcal{S}_n, \; \forall i \in \{1, \ldots, n\}, \; \text{sgn}(\sigma) \prod_{i=1}^n \mathbf{T}_{i,\sigma(i)} \neq 0. \tag{42}$$

This is equivalent to

$$\exists \sigma \in \mathcal{S}_n, \; \forall i \in \{1, \ldots, n\}, \; \mathbf{T}_{i,\sigma(i)} \neq 0. \tag{43}$$

Then we can see that this $\sigma$ is in the support of $\mathbf{T}$, which implies that

$$\forall j \in \{1, \ldots, n\}, \; \sigma(j) \in \mathcal{T}_{j,:}. \tag{44}$$

Together with Eq. (40), it follows that

$$\forall (i, j) \in \mathcal{A}, (i, \sigma(j)) \in \{i\} \times \mathcal{T}_{j,:} \subset \hat{\mathcal{A}}. \tag{45}$$

Denote

$$\sigma(\mathcal{A}) = \{(i, \sigma(j)) \mid (i, j) \in \mathcal{A}\}. \tag{46}$$

Then we have

$$\sigma(\mathcal{A}) \subset \hat{\mathcal{A}}. \tag{47}$$

According to Assumption iii, we can see that

$$|\hat{\mathcal{A}}| \leq |\mathcal{A}| = |\sigma(\mathcal{A})|. \tag{48}$$

Together with Eq. (47), we have

$$\sigma(\mathcal{A}) = \hat{\mathcal{A}}. \tag{49}$$

Suppose $\mathbf{T} \neq \mathbf{D}\mathbf{P}$, there must exists $j_1$ and $j_2$ such that

$$\exists j_1 \neq j_2, \; \mathcal{T}_{j_1,:} \cap \mathcal{T}_{j_2,:} \neq \emptyset. \tag{50}$$

Besides, consider $j_3 \in \{1, \ldots, n\}$ such that

$$\sigma(j_3) \in \mathcal{T}_{j_1,:} \cap \mathcal{T}_{j_2,:}. \tag{51}$$

Because $j_1 \neq j_2$, we can assume $j_3 \neq j_1$ without loss of generality. By Assumption iv, there exists $\mathcal{C}_{j_1} \ni j_1$ such that $\bigcap_{i \in \mathcal{C}_{j_1}} \mathcal{A}_{i,:} = \{j_1\}$. Because

$$j_3 \notin \{j_1\} = \bigcap_{i \in \mathcal{C}_{j_1}} \mathcal{A}_{i,:}, \tag{52}$$

there must exists $i_3 \in \mathcal{C}_{j_1}$ such that

$$j_3 \notin \mathcal{A}_{i_3,:}. \tag{53}$$

Because $j_1 \in \mathcal{A}_{i_3,:}$, we have $(i_3, j_1) \in \mathcal{A}$. Then according to Eq. (40), we have the following equation

$$\{i_3\} \times \mathcal{T}_{j_1,:} \subset \hat{\mathcal{A}}. \tag{54}$$

Notice that $\sigma(j_3) \in \mathcal{T}_{j_1,:} \cap \mathcal{T}_{j_2,:}$ implies

$$(i_3, \sigma(j_3)) \in \{i_3\} \times \mathcal{T}_{j_1,:}. \tag{55}$$

Then by Eqs. (55) and (54), we have

$$(i_3, \sigma(j_3)) \in \hat{\mathcal{A}}, \tag{56}$$

which implies $(i_3, j_3) \in \mathcal{A}$ by Eqs. (49) and (46), contradicting Eq. (53). Thus, we prove by contradiction that $\mathbf{T} = \mathbf{DP}$, which is the goal (i.e., Eq. (37)). $\qquad\square$

### A.4 Proof of Theorem 2

**Theorem 2.** *Let the observed data be sampled from a linear ICA model defined in Eqs. (1) and (3) with Gaussian sources. Differently, the number of observed variables (denoted as $m$) could be larger than that of the sources $n$, i.e., $m \geq n$. Suppose the following assumptions hold:*

  i. *The nonzero coefficients of the mixing matrix $\mathbf{A}$ are randomly drawn from a distribution that is absolutely continuous with respect to Lebesgue measure.*

  ii. *The estimated mixing matrix $\hat{\mathbf{A}}$ has the minimal $L_0$ norm during estimation.*

  iii. *(Structural Sparsity) Given $\mathcal{C} \subseteq \{1, 2, \ldots, n\}$ where $|\mathcal{C}| > 1$, let $\mathbf{A}_\mathcal{C} \in \mathbb{R}^{m \times |\mathcal{C}|}$ represents a submatrix of $\mathbf{A} \in \mathbb{R}^{m \times n}$ consisting of columns with indices $\mathcal{C}$. Then, for all $k \in \mathcal{C}$, we have*

$$\left| \bigcup_{k' \in \mathcal{C}} \mathrm{supp}(\mathbf{A}_{k'}) \right| - \mathrm{rank}(\mathrm{overlap}(\mathbf{A}_\mathcal{C})) > |\mathrm{supp}(\mathbf{A}_k)|.$$

*Then $\hat{\mathbf{A}} = \mathbf{ADP}$ with probability one, where $\mathbf{D}$ is a diagonal matrix and $\mathbf{P}$ is a column permutation matrix.*

*Proof.* Because of Assumptions ( ii), we consider the following combinatorial optimization

$$\hat{\mathbf{U}} \;:=\; \underset{\substack{\mathbf{U} \in \mathbb{R}^{s \times s}; \\ \mathbf{U}\mathbf{U}^\top = \mathbf{I}_s}}{\arg\min} \; \|\mathbf{A}\mathbf{U}\|_0, \tag{57}$$

where $\mathbf{A}$ is the true mixing matrix and $\hat{\mathbf{U}}$ denotes the rotation matrix corresponding to the solution of the optimization problem. Let $\hat{\mathbf{A}} = \mathbf{A}\hat{\mathbf{U}}$.

Suppose $\hat{\mathbf{A}} \neq \mathbf{ADP}$, then $\hat{\mathbf{U}} \neq \mathbf{DP}$. This implies that there exists some $j' \in \{1, \ldots, s\}$ and its corresponding set of row indices $\mathcal{I}_{j'}$ ($|\mathcal{I}_{j'}| > 1$), such that $\hat{\mathbf{U}}_{i,j'} \neq 0$ for all $i \in \mathcal{I}_{j'}'$, and $\hat{\mathbf{U}}_{i,j'} = 0$ for all $i \notin \mathcal{I}_{j'}$. Because $\hat{U}$ is invertible and has full row rank, there exists one row index $i'$ in $\mathcal{I}_{j'}$ that uniquely correspond to $j'$, in order to avoid linear dependence among columns. Let $\hat{\mathbf{U}} := \left[ \hat{\mathbf{U}}_1 \cdots \hat{\mathbf{U}}_s \right]$, we have

$$\left\| \hat{\mathbf{A}}_{j'} \right\|_0 = \left\| \mathbf{A}\hat{\mathbf{U}}_{j'} \right\|_0 = \left\| \sum_{i \in \mathcal{I}_{j'}} \mathbf{A}_i \hat{\mathbf{U}}_{i,j'} \right\|_0. \tag{58}$$

Let $\mathbf{A}_{\mathcal{I}_{j'}} \in \mathbb{R}^{m \times |\mathcal{I}_{j'}|}$ represents a submatrix of $\mathbf{A}$ consisting of columns with indices $\mathcal{I}_{j'}$. Note that with a slight abuse of notation, $\mathbf{A}_i$ denotes $i$-th column of the matrix $\mathbf{A}$. According to Assumptions (i, iii), since $|\mathcal{I}_{j'}| > 1$ and $\hat{\mathbf{U}}_{i,j'} \neq 0$, we have

$$\left\| \sum_{i \in \mathcal{I}_{j'}} \mathbf{A}_i \hat{\mathbf{U}}_{i,j'} \right\|_0 \geq \left| \bigcup_{i \in \mathcal{I}_{j'}} \mathrm{supp}(\mathbf{A}_i) \right| - \mathrm{rank}(\mathrm{overlap}(\mathbf{A}_{\mathcal{I}_{j'}})) > |\mathrm{supp}(\mathbf{A}_{i'})|, \quad (59)$$

where $\mathrm{overlap}(\cdot)$ is defined as Defn. 4. Term $\mathrm{rank}(\mathrm{overlap}(\mathbf{A}_{\mathcal{I}_{j'}}))$ represents the maximal number of rows, in which all non-zero entries can be possibly cancelled out by the linear combination $\sum_{i \in \mathcal{I}_{j'}} \mathbf{A}_i$. Assumption i rules out a specific set of parameters that leads to a violation of that, e.g., two columns of $\mathbf{A}$ are identical in terms of element values and support. So it follows that

$$\left\| \sum_{i \in \mathcal{I}_{j'}} \mathbf{A}_i \hat{\mathbf{U}}_{i,j'} \right\|_0 > |\mathrm{supp}(\mathbf{A}_{i'})| = \|\mathbf{A}_{i'}\|_0 = \left\| \mathbf{A}_{i'} \hat{\mathbf{U}}_{i',j'} \right\|_0. \quad (60)$$

Then we can construct $\tilde{\mathbf{U}} := \begin{bmatrix} \tilde{\mathbf{U}}_1 \cdots \tilde{\mathbf{U}}_s \end{bmatrix}$. First, we set $\tilde{\mathbf{U}}_{i',j'}$ as a unique non-zero entry in column $\tilde{\mathbf{U}}_{j'}$. For simplicity, we can just set $\tilde{\mathbf{U}}_{i',j'} = 1$. For other column $\tilde{\mathbf{U}}_j$, where $j \neq j'$ and $\hat{\mathbf{U}}_{i,j} \neq 0$, we set $\tilde{\mathbf{U}}_{i,j} = 1$. Therefore

$$\begin{cases} \left\| \mathbf{A} \hat{\mathbf{U}}_j \right\|_0 > \left\| \mathbf{A} \tilde{\mathbf{U}}_j \right\|_0, & j = j', \\ \left\| \mathbf{A} \hat{\mathbf{U}}_j \right\|_0 = \left\| \mathbf{A} \tilde{\mathbf{U}}_j \right\|_0, & j \neq j'. \end{cases} \quad (61)$$

Since Assumption iii covers all columns, Eq. (60) holds for any $j' \in \{1, \ldots, s\}$. If there are multiple columns of $\hat{U}$ with more than one nonzero entry, we derive Eq. (60) for each of them. We denote the set of different target column indices $j'$ as $\mathbf{J}$. For $j \in \mathbf{J}$, we set $\tilde{\mathbf{U}}_{i_j,j} = 1$, where $i_j$ is the unique index of the corresponding non-zero entry in column $\hat{U}_j$. For other column $\tilde{\mathbf{U}}_j$, where $j \notin \mathbf{J}$ and $\hat{\mathbf{U}}_{i,j} \neq 0$, we set $\tilde{\mathbf{U}}_{i,j} = 1$. Then we have

$$\begin{cases} \left\| \mathbf{A} \hat{\mathbf{U}}_j \right\|_0 > \left\| \mathbf{A} \tilde{\mathbf{U}}_j \right\|_0, & j \in \mathbf{J}, \\ \left\| \mathbf{A} \hat{\mathbf{U}}_j \right\|_0 = \left\| \mathbf{A} \tilde{\mathbf{U}}_j \right\|_0, & j \notin \mathbf{J}. \end{cases} \quad (62)$$

As noted previously, every column index $j$ corresponds to a unique row index. $\tilde{\mathbf{U}}$ is a permutation matrix and $\tilde{\mathbf{U}}\tilde{\mathbf{U}}^\top = \mathbf{I}_s$. It then follows that $\left\| \mathbf{A}\hat{\mathbf{U}} \right\|_0 > \left\| \mathbf{A}\tilde{\mathbf{U}} \right\|_0$, which contradicts the definition of $\hat{\mathbf{U}}$. $\qquad \square$

## A.5 Proof of Theorem 3

**Theorem 3.** *Given a nonlinear ICA model defined in Eqs. (1) and (2), where $\mathbf{f}$ is the true mixing function. Consider $\hat{\mathbf{f}} = \mathbf{f} \circ \mathbf{G}^{-1} \circ \mathbf{U} \circ \mathbf{G}$, where $\mathbf{G}$ denotes an invertible Gaussianization[3] that maps the distribution to an standard isotropic (rotation-invariant) Gaussian, $\mathbf{U}$ denotes a rotation, and $\mathbf{G}^{-1}$ maps the distribution back to that before applying $\hat{\mathbf{U}} \circ \mathbf{G}$. If Assumptions i, ii and iii of Thm. 2 are satisfied by replacing $\mathbf{A}$ with $\mathbf{J}_{\mathbf{f}}(\mathbf{s})$ and $\hat{\mathbf{A}}$ with $\mathbf{J}_{\hat{\mathbf{f}}}(\mathbf{s})$, then function $\mathbf{h} := \hat{\mathbf{f}}^{-1} \circ \mathbf{f}$ is a composition of a component-wise invertible transformation and a permutation with probability one.*

---

[3]One example is described in (Gresele et al., 2021), i.e., a composition of the element-wise CDFs of a smooth factorised density and a Gaussian, respectively.

*Proof.* Let $\mathbf{D}(\mathbf{s})$ represents a diagonal matrix and $\mathbf{P}$ represent a permutation matrix. By using the chain rule repeatedly, we write $\hat{\mathbf{f}} = \mathbf{f} \circ \mathbf{h}(\mathbf{s})$ equivalently as

$$
\begin{aligned}
\mathbf{J}_{\hat{\mathbf{f}}}(\hat{\mathbf{s}}) &= \mathbf{J}_{\mathbf{f} \circ \mathbf{h}}(\mathbf{h}(\mathbf{s})) \\
&= \mathbf{J}_{\mathbf{f} \circ \mathbf{g}^{-1} \circ \mathbf{P}^{-1}}(\mathbf{P}\mathbf{g}(\mathbf{s})) \\
&= \mathbf{J}_{\mathbf{f} \circ \mathbf{g}^{-1}}\left(\mathbf{P}^{-1}\mathbf{P}\mathbf{g}(\mathbf{s})\right) \mathbf{J}_{\mathbf{P}^{-1}}(\mathbf{P}\mathbf{g}(\mathbf{s})) \\
&= \mathbf{J}_{\mathbf{f} \circ \mathbf{g}^{-1}}(\mathbf{g}(\mathbf{s}))\mathbf{J}_{\mathbf{P}^{-1}}(\mathbf{P}\mathbf{g}(\mathbf{s})) \\
&= \mathbf{J}_{\mathbf{f}}\left(\mathbf{g}^{-1}\mathbf{g}(\mathbf{s})\right) \mathbf{J}_{\mathbf{g}^{-1}}(\mathbf{g}(\mathbf{s}))\mathbf{J}_{\mathbf{P}^{-1}}(\mathbf{P}\mathbf{g}(\mathbf{s})) \\
&= \mathbf{J}_{\mathbf{f}}(\mathbf{s})\mathbf{D}(\mathbf{s})\mathbf{P},
\end{aligned}
\tag{63}
$$

where $\mathbf{g}$ is an invertible element-wise function. Thus our goal is equivalent to show that

$$
\mathbf{J}_{\hat{\mathbf{f}}}(\hat{\mathbf{s}}) = \mathbf{J}_{\mathbf{f}}(\mathbf{s})\mathbf{D}(\mathbf{s})\mathbf{P}.
\tag{64}
$$

We prove it by contrapositive.

Because $\hat{\mathbf{f}} = \mathbf{f} \circ \mathbf{G}^{-1} \circ \mathbf{U} \circ \mathbf{G}$, we write

$$
\begin{aligned}
\mathbf{J}_{\hat{\mathbf{f}}}(\mathbf{s}) &= \mathbf{J}_{\mathbf{f} \circ \mathbf{G}^{-1} \circ \hat{\mathbf{U}} \circ \mathbf{G}}(\mathbf{s}) \\
&= \mathbf{J}_{\mathbf{f} \circ \mathbf{G}^{-1} \circ \hat{\mathbf{U}}}(\mathbf{G}(\mathbf{s}))\mathbf{J}_{\mathbf{G}}(\mathbf{s}) \\
&= \mathbf{J}_{\mathbf{f} \circ \mathbf{G}^{-1}}(\hat{\mathbf{U}}\mathbf{G}(\mathbf{s}))\mathbf{J}_{\hat{U}}(\mathbf{G}(\mathbf{s}))\mathbf{J}_{\mathbf{G}}(\mathbf{s}) \\
&= \mathbf{J}_{\mathbf{f}}(\mathbf{G}^{-1}\hat{\mathbf{U}}\mathbf{G}(\mathbf{s}))\mathbf{J}_{\mathbf{G}^{-1}}(\hat{\mathbf{U}}\mathbf{G}(\mathbf{s}))\mathbf{J}_{\hat{U}}(\mathbf{G}(\mathbf{s}))\mathbf{J}_{\mathbf{G}}(\mathbf{s}) \\
&= \mathbf{J}_{\mathbf{f}}(\mathbf{s})\mathbf{J}_{\mathbf{G}^{-1}}(\hat{\mathbf{U}}\mathbf{G}(\mathbf{s}))\mathbf{J}_{\hat{U}}(\mathbf{G}(\mathbf{s}))\mathbf{J}_{\mathbf{G}}(\mathbf{s}) \\
&= \mathbf{J}_{\mathbf{f}}(\mathbf{s})\mathbf{D}_1(\mathbf{s})\hat{\mathbf{U}}\mathbf{D}_2(\mathbf{s}),
\end{aligned}
\tag{65}
$$

where we have used the chain rule repeatedly. Because $\mathbf{G}$ is an invertible element-wise transformation, $\mathbf{D}_1(\mathbf{s})$ and $\mathbf{D}_2(\mathbf{s})$ are both diagonal matrices.

Because Assumption ii of Thm. 2 is satisfied for $\mathbf{J}_{\mathbf{f}}(\mathbf{s})$, we consider the following combinatorial optimization problem

$$
\hat{\mathbf{U}} := \underset{\substack{\mathbf{U} \in \mathbb{R}^{s \times s}; \\ \mathbf{U}\mathbf{U}^T = \mathbf{I}_s}}{\arg\min} \|\mathbf{J}_{\mathbf{f}}(\mathbf{s})\mathbf{U}\|_0.
\tag{66}
$$

Let $\bar{\mathbf{U}} = \mathbf{D}_1(\mathbf{s})\hat{\mathbf{U}}\mathbf{D}_2(\mathbf{s})$. Because $\mathbf{J}_{\hat{\mathbf{f}}}(\mathbf{s}) = \mathbf{J}_{\mathbf{f}}(\mathbf{s})\bar{\mathbf{U}}$, then if $\mathbf{J}(\hat{\mathbf{f}}) = \mathbf{J}_{\mathbf{f}}(\mathbf{s})\mathbf{D}(\mathbf{s})\mathbf{P}$, we have $\bar{\mathbf{U}} = \mathbf{D}(\mathbf{s})\mathbf{P}$. Thus, for every $j \in \{1, \ldots, s\}$, there exists a corresponding $i'$, such that $\bar{\mathbf{U}}_{i',j} \neq 0$, and $\bar{\mathbf{U}}_{i,j} = 0$ for all $i \neq i'$. Because the columns of the matrix of an orthogonal transformation form an orthogonal set, columns of $\bar{\mathbf{U}}$ are linearly independent. Thus, different $j$ cannot correspond to the same $i'$, otherwise it is possible for these columns to be linearly dependent.

Suppose $\mathbf{J}_{\hat{\mathbf{f}}}(\mathbf{s}) \neq \mathbf{J}_{\mathbf{f}}(\mathbf{s})\mathbf{P}$, then $\bar{\mathbf{U}} \neq \mathbf{P}$. There exists $j' \in \{1, \ldots, s\}$ and its corresponding set of row indices $\mathcal{I}_{j'}$ ($|\mathcal{I}_{j'}| > 1$), such that $\bar{\mathbf{U}}_{i,j'} \neq 0$ for all $i \in \mathcal{I}'_j$, and $\bar{\mathbf{U}}_{i,j'} = 0$ for all $i \notin \mathcal{I}_{j'}$. Similarly, there exists one row index $i'$ in $\mathcal{I}_{j'}$ that uniquely correspond to $j'$, in order to avoid linear dependence among columns. Let $\bar{\mathbf{U}} := [\bar{\mathbf{U}}_1 \cdots \bar{\mathbf{U}}_s]$, we have

$$
\left\|\mathbf{J}_{\hat{\mathbf{f}}}(\mathbf{s})_{j'}\right\|_0 = \left\|\mathbf{J}_{\mathbf{f}}(\mathbf{s})\bar{\mathbf{U}}_{j'}\right\|_0 = \left\|\sum_{i \in \mathcal{I}_{j'}} \mathbf{J}_{\mathbf{f}}(\mathbf{s})_i \bar{\mathbf{U}}_{i,j'}\right\|_0.
\tag{67}
$$

Let $\mathbf{J}_{\mathbf{f}}(\mathbf{s})_{\mathcal{I}_{j'}} \in \mathbb{R}^{m \times |\mathcal{I}_{j'}|}$ represents a submatrix of $\mathbf{J}_{\mathbf{f}}(\mathbf{s})$ consisting of columns with indices $\mathcal{I}_{j'}$. Note that with a slight abuse of notation, $\mathbf{J}_{\mathbf{f}}(\mathbf{s})_i$ denotes $i$-th column of the matrix $\mathbf{J}_{\mathbf{f}}(\mathbf{s})$. Because Assumptions (i, iii) of Thm. 2 hold for $\mathbf{J}_{\mathbf{f}}(\mathbf{s})$, with $|\mathcal{I}_{j'}| > 1$ and $\bar{\mathbf{U}}_{i,j'} \neq 0$, we have

$$
\left\|\sum_{i \in \mathcal{I}_{j'}} \mathbf{J}_{\mathbf{f}}(\mathbf{s})_i \bar{\mathbf{U}}_{i,j'}\right\|_0 \geq \left|\bigcup_{i \in \mathcal{I}_{j'}} \text{supp}(\mathbf{J}_{\mathbf{f}}(\mathbf{s})_i)\right| - \text{rank}(\text{overlap}(\mathbf{J}_{\mathbf{f}}(\mathbf{s})_{\mathcal{I}_{j'}})) > |\text{supp}(\mathbf{J}_{\mathbf{f}}(\mathbf{s})_{i'})|, \tag{68}
$$

where $\text{overlap}(\cdot)$ is defined as Defn. 4. Note that here we slightly abuse the notation $\text{overlap}(\cdot)$ to make it apply for the matrix-valued function $\mathbf{J_f(s)}$. Term $\text{rank}(\text{overlap}(\mathbf{J_f(s)}_{\mathcal{I}_{j'}}))$ generally represents the maximal number of rows, in which all non-zero entries can be cancelled out by the linear combination $\sum_{i \in \mathcal{I}_{j'}} \mathbf{J_f(s)}_i$. Also with a slight abuse of the notation, Assumption i of Thm. 2 rules out a specific set of parameters that leads to a violation of that, e.g., two columns of $\mathbf{J_f(s)}$ are identical in terms of element values and support. So it follows that

$$\left\| \sum_{i \in \mathcal{I}_{j'}} \mathbf{J_f(s)}_i \bar{\mathbf{U}}_{i,j'} \right\|_0 > |\text{supp}(\mathbf{J_f(s)}_{i'})| = \left\| \mathbf{J_f(s)}_{i'} \right\|_0 = \left\| \mathbf{J_f(s)}_{i'} \bar{\mathbf{U}}_{i',j'} \right\|_0. \tag{69}$$

Then we can construct $\tilde{\mathbf{U}} := \left[ \tilde{\mathbf{U}}_1 \cdots \tilde{\mathbf{U}}_s \right]$. First, we set $\tilde{\mathbf{U}}_{i',j'}$ as a unique non-zero entry in column $\tilde{\mathbf{U}}_{j'}$. For simplicity, we can just set $\tilde{\mathbf{U}}_{i',j'} = 1$. For other column $\tilde{\mathbf{U}}_j$, where $j \neq j'$ and $\bar{\mathbf{U}}_{i,j} \neq 0$, we set $\tilde{\mathbf{U}}_{i,j} = 1$. Therefore

$$\begin{cases} \left\| \mathbf{J_f(s)} \bar{\mathbf{U}}_j \right\|_0 > \left\| \mathbf{J_f(s)} \tilde{\mathbf{U}}_j \right\|_0, & j = j', \\ \left\| \mathbf{J_f(s)} \bar{\mathbf{U}}_j \right\|_0 = \left\| \mathbf{J_f(s)} \tilde{\mathbf{U}}_j \right\|_0, & j \neq j'. \end{cases} \tag{70}$$

Since Assumption iii of Thm. 2 covers all columns, Eq. (69) holds for any $j' \in \{1, \ldots, s\}$. If there are multiple columns of $\bar{U}$ with more than one nonzero entry, we derive Eq. (69) for each of them. We denote the set of different target column indices $j'$ as $\mathbf{J}$. For $j \in \mathbf{J}$, we set $\tilde{\mathbf{U}}_{i_j,j} = 1$, where $i_j$ is the unique index of the corresponding non-zero entry in column $\hat{U}_j$. For other column $\tilde{\mathbf{U}}_j$, where $j \notin \mathbf{J}$ and $\bar{\mathbf{U}}_{i,j} \neq 0$, we set $\tilde{\mathbf{U}}_{i,j} = 1$. Then we have

$$\begin{cases} \left\| \mathbf{J_f(s)} \bar{\mathbf{U}}_j \right\|_0 > \left\| \mathbf{J_f(s)} \tilde{\mathbf{U}}_j \right\|_0, & j \in \mathbf{J}, \\ \left\| \mathbf{J_f(s)} \bar{\mathbf{U}}_j \right\|_0 = \left\| \mathbf{J_f(s)} \tilde{\mathbf{U}}_j \right\|_0, & j \notin \mathbf{J}. \end{cases} \tag{71}$$

As noted previously, every column index $j$ corresponds to a unique row index $i$. $\tilde{\mathbf{U}}$ is a permutation matrix and $\tilde{\mathbf{U}}\tilde{\mathbf{U}}^\top = \mathbf{I}_s$. It then follows that $\left\| \mathbf{J_f(s)} \bar{\mathbf{U}} \right\|_0 > \left\| \mathbf{J_f(s)} \tilde{\mathbf{U}} \right\|_0$, which contradicts the definition of $\bar{\mathbf{U}}$. $\qquad\square$

### A.6 Proof of Proposition 2

**Proposition 2.** *Let the observed data be sampled from a nonlinear ICA model as defined in Eqs. (1), (2), and (7). Suppose the following assumptions hold:*

  i. *(Independent Influences): The influence of each source on the observed variables is independent of each other, i.e., each partial derivative $\partial \mathbf{f}/\partial s_i$ is independent of the other sources and their influences in a non-statistical sense.*

  ii. *The mixing function $\mathbf{f}$ and its inverse are twice differentiable.*

  iii. *The Jacobian determinant of mixing function can be factorized as $\det(\mathbf{J_f(s)}) = \prod_{i=1}^n y_i(s_i)$, where $y_i$ is a function that depends only on $s_i$. Note that volume-preserving transformation is a special case when $y_i(s_i) = 1, i = 1, \ldots, n$.*

  iv. *During estimation, the columns of the Jacobian of the estimated unmixing function are regularized to be mutually orthogonal and with equal euclidean norm.*

*Then $\mathbf{h} := \hat{\mathbf{f}}^{-1} \circ \mathbf{f}$ is a composition of a component-wise invertible transformation and a permutation.*

*Proof.* We first restate the setup here. Consider the following data-generating process of nonlinear ICA

$$p_\mathbf{s}(\mathbf{s}) = \prod_{i=1}^{n} p_{s_i}(s_i), \tag{72}$$

$$\mathbf{x} = \mathbf{f}(\mathbf{s}), \tag{73}$$

where $\mathbf{x}$ denotes the observed random vector, $\mathbf{s}$ is the latent random vector representing the marginally independent sources, $p_{s_i}$ is the marginal PDF of $s_i$, $p_\mathbf{s}$ is the joint PDF, and $\mathbf{f} : \mathbf{s} \to \mathbf{x}$ denotes a nonlinear mixing function. We preprocess the observational data by centering the columns of the Jacobian of the mixing function. This could be achieved by applying a fixed transformation on the observational data that is equivalent to left multiplying the Jacobian of the mixing function with the centering matrix $\mathbf{U} := \mathbf{I}_n - (1/n)\mathbf{1}\mathbf{1}^\top$, where $\mathbf{I}_n$ is the identity matrix of size $n$ and $\mathbf{1}$ is the $n$-dimensional vector of all ones. After centering, according to the assumption of independent influences, the columns of the Jacobian matrix are uncorrelated and of zero-means, and thus orthogonal to each other [4]. Since the preprocessing transformation $\mathbf{U}$ is fixed w.r.t. no variables other than $n$, which is the fixed number of dimensions, it could always be recovered after reconstruction.

We assume that the distribution of the estimated sources $\hat{\mathbf{s}}$ follows a factorial multivariate Gaussian:

$$p_{\hat{\mathbf{s}}}(\hat{\mathbf{s}}) = \prod_{i=1}^{n} \frac{1}{Z_i} \exp\left(-\theta_{i,1}\hat{s}_i - \theta_{i,2}\hat{s}_i^2\right), \tag{74}$$

where $Z_i > 0$ is a constant. The sufficient statistics $\theta_{i,1} = -\frac{\mu_i}{\sigma_i^2}$ and $\theta_{i,2} = \frac{1}{2\sigma_i^2}$ are assumed to be linearly independent. We set the variances $\sigma_i^2$ to be distinct without loss of generality, which can be placed as a constraint during the estimation process as the sufficient statistics of the estimated distribution are trainable.

Our goal here is to show that function $\mathbf{h} = \hat{\mathbf{f}}^{-1} \circ \mathbf{f}$ is a composition of a permutation and a component-wise invertible transformation of sources. By using chain rule repeatedly, we write the Jacobian of function $\mathbf{h} := \hat{\mathbf{f}}^{-1} \circ \mathbf{f}$ as

$$\begin{aligned}
\mathbf{J_h}(\mathbf{s}) &= \mathbf{J}_{\hat{\mathbf{f}}^{-1} \circ \mathbf{f}}(\mathbf{s}) \\
&= \mathbf{J}_{\hat{\mathbf{f}}^{-1}}(\mathbf{f}(\mathbf{s}))\mathbf{J_f}(\mathbf{s}) \\
&= \mathbf{J}_{\hat{\mathbf{f}}^{-1}}(\mathbf{x})\mathbf{J_f}(\mathbf{s}) \\
&= \hat{\mathbf{O}}(\mathbf{x})\hat{\lambda}(\mathbf{x})\mathbf{O}(\mathbf{s})\mathbf{D}(\mathbf{s}) \\
&= \hat{\lambda}(\mathbf{x})\hat{\mathbf{O}}(\mathbf{x})\mathbf{O}(\mathbf{s})\mathbf{D}(\mathbf{s}),
\end{aligned} \tag{75}$$

where $\hat{\lambda}(\mathbf{x})$ is a scalar, $\hat{\mathbf{O}}(\mathbf{x})$ and $\mathbf{O}(\mathbf{s})$ represent the corresponding orthogonal matrices, and $\mathbf{D}(\mathbf{s})$ denotes the corresponding diagonal matrix. The fourth equality follows from assumption i.

Since $\mathbf{J_h}(\mathbf{s}) = \hat{\lambda}(\mathbf{x})\hat{\mathbf{O}}(\mathbf{x})\mathbf{O}(\mathbf{s})\mathbf{D}(\mathbf{s})$ by Eq. (75) and $\hat{\mathbf{O}}(\mathbf{x})\mathbf{O}(\mathbf{s})$ is an orthogonal matrix, the columns of $\mathbf{J_h}(\mathbf{s})$ are orthogonal to each other, i.e.,

$$\mathbf{J_h}(\mathbf{s})_{:,j}^\top \mathbf{J_h}(\mathbf{s})_{:,k} = 0, \quad \forall j \neq k, \tag{76}$$

where $\mathbf{J_h}(\mathbf{s})_{:,j}$ is the $j$-th column of matrix $\mathbf{J_h}(\mathbf{s})$. This can be rewritten as

$$\sum_{i=1}^{n} \left(\frac{\partial h_i}{\partial s_j}\frac{\partial h_i}{\partial s_k}\right) = 0, \quad \forall j \neq k. \tag{77}$$

By assumption, the components of the true sources $\mathbf{s}$ are mutually independent, while the estimated sources $\hat{\mathbf{s}}$ follows a multivariate Gaussian distribution:

$$\begin{aligned}
p_\mathbf{s}(\mathbf{s}) &= \prod_{i=1}^{n} p_{\mathbf{s}_i}(s_i), \\
p_{\hat{\mathbf{s}}}(\hat{\mathbf{s}}) &= \prod_{i=1}^{n} \frac{1}{Z_i} \exp\left(-\theta_{i,1}\hat{s}_i - \theta_{i,2}\hat{s}_i^2\right),
\end{aligned} \tag{78}$$

---

[4]Similarly, Gresele et al. (2021) directly formalize the notion of Independent Mechanism Analysis as orthogonality of the columns of the Jacobian.

where $Z_i > 0$ is a constant. The sufficient statistics $\theta_{i,1}$ and $\theta_{i,2}$ are assumed to be linearly independent.

Applying the change of variable rule, we have $p_{\mathbf{s}}(\mathbf{s}) = p_{\hat{\mathbf{s}}}(\hat{\mathbf{s}})|\det(\mathbf{J_h}(\mathbf{s}))|$, which, by plugging in Eq. (78) and taking the logarithm on both sides, yields

$$\sum_{i=1}^{n} \log p_{s_i}(s_i) - \log|\det(\mathbf{J_h}(\mathbf{s}))| = -\sum_{i=1}^{n} \left(\theta_{i,1}h_i(\mathbf{s}) + \theta_{i,2}h_i(\mathbf{s})^2 + \log Z_i\right). \qquad (79)$$

According to assumption iii, we have the factorization $\det(\mathbf{J_h}(\mathbf{s})) = \prod_{i=1}^{n} y_i(s_i)$, where $y_i$ is a function that depends only on $s_i$. This implies that

$$\sum_{i=1}^{n} \left(\log p_{s_i}(s_i) - |y_i(s_i)|\right) = -\sum_{i=1}^{n} \left(\theta_{i,1}h_i(\mathbf{s}) + \theta_{i,2}h_i(\mathbf{s})^2 + \log Z_i\right). \qquad (80)$$

Then, based on Lemma 1 by Yang et al. (2022), we have the following equation:

$$\sum_{i=1}^{n} \left(\frac{1}{\sigma_i^2} \frac{\partial h_i}{\partial s_j} \frac{\partial h_i}{\partial s_k}\right) = 0, \quad \forall j \neq k. \qquad (81)$$

Specifically, Eq. (80) is similar to Eq. (17') and Eq. (17) in Yang et al. (2022). Eq. (81) is similar to Eq. (20) in Yang et al. (2022). The only difference is that there is no auxiliary variable in Eqs. (80) and (81), which does not influence the deriviation of Lemma 1 by Yang et al. (2022). The assumptions of the lemma are satisfied here, with the alternation that Eq. (12) in Yang et al. (2022) is weakened to Eq. (17') in Yang et al. (2022), which does not influence the deriviation according to Appx. B in Yang et al. (2022). Thus, based on Lemma 1 in (Yang et al., 2022) but without auxiliary variables, we could obtain Eq. (81).

Then, we define the following matrices:

$$\begin{aligned}
\mathbf{\Lambda}_1 &= \mathrm{diag}\left(\frac{1}{\sigma_1^2}, \cdots, \frac{1}{\sigma_n^2}\right), \\
\mathbf{\Lambda}_2 &= \mathbf{I}_n, \\
\mathbf{\Sigma}(1, \mathbf{s}) &= \mathrm{diag}\left(\sum_{i=1}^{n} \frac{1}{\sigma_i^2}\left(\frac{\partial h_i}{\partial s_1}\right)^2, \cdots, \sum_{i=1}^{n} \frac{1}{\sigma_i^2}\left(\frac{\partial h_i}{\partial s_n}\right)^2\right), \\
\mathbf{\Sigma}(2, \mathbf{s}) &= \mathrm{diag}\left(\sum_{i=1}^{n} \left(\frac{\partial h_i}{\partial s_1}\right)^2, \cdots, \sum_{i=1}^{n} \left(\frac{\partial h_i}{\partial s_n}\right)^2\right).
\end{aligned} \qquad (82)$$

We rewrite Eq. (81) and Eq. (77) as

$$\begin{cases} \mathbf{J_h}(\mathbf{s})^{\top}\mathbf{\Lambda}_1\mathbf{J_h}(\mathbf{s}) = \mathbf{\Sigma}(1, \mathbf{s}), \\ \mathbf{J_h}(\mathbf{s})^{\top}\mathbf{\Lambda}_2\mathbf{J_h}(\mathbf{s}) = \mathbf{\Sigma}(2, \mathbf{s}). \end{cases} \qquad (83)$$

Because all elements in matrices defined in Eq. (82) are positive, we can take sqaure roots on both sides of equations in Eq. (83), yielding

$$\begin{cases} \mathbf{\Lambda}_1^{1/2}\mathbf{J_h}(\mathbf{s}) = \mathbf{O}_1\mathbf{\Sigma}(1, \mathbf{s})^{1/2}, \\ \mathbf{\Lambda}_2^{1/2}\mathbf{J_h}(\mathbf{s}) = \mathbf{O}_2\mathbf{\Sigma}(2, \mathbf{s})^{1/2}, \end{cases} \qquad (84)$$

where $\mathbf{O}_1$ and $\mathbf{O}_2$ are the corresponding orthogonal matrices ($\mathbf{O}_1^{\top}\mathbf{O}_1 = \mathbf{O}_2^{\top}\mathbf{O}_2 = \mathbf{I}_n$) and $\mathbf{\Lambda}^{1/2}$ denotes the matrix by applying entry-wise square root to the entries of diagonal matrix $\mathbf{\Lambda}$. Then we rewrite Eq. (84) as

$$\begin{cases} \mathbf{J_h}(\mathbf{s}) = \mathbf{\Lambda}_1^{-1/2}\mathbf{O}_1\mathbf{\Sigma}(1, \mathbf{s})^{1/2}, \\ \mathbf{J_h}(\mathbf{s}) = \mathbf{\Lambda}_2^{-1/2}\mathbf{O}_2\mathbf{\Sigma}(2, \mathbf{s})^{1/2}. \end{cases} \qquad (85)$$

By reformulating, we have

$$\mathbf{\Sigma}(1, \mathbf{s})^{1/2}\mathbf{\Sigma}(2, \mathbf{s})^{-1/2} = \mathbf{O}_1^{-1}\mathbf{\Lambda}_1^{1/2}\mathbf{\Lambda}_2^{-1/2}\mathbf{O}_2. \qquad (86)$$

We consider both sides of Eq. (86) as a singular value decomposition of the LHS. Since $\sigma_j \neq \sigma_k, \forall j \neq k$, elements in $\mathbf{\Lambda}_1^{1/2}\mathbf{\Lambda}_2^{-1/2}$ are distinct. Therefore, non-zero elements in $\mathbf{\Lambda}_1^{1/2}\mathbf{\Lambda}_2^{-1/2}$ are also distinct because $\mathbf{\Lambda}_2$ is an identity matrix.

Because the mixing function is assumed to be invertible, the Jacobian of the mixing function $\mathbf{J}_{\mathbf{f}}(\mathbf{s}) = \mathbf{O}(\mathbf{s})\mathbf{D}(\mathbf{s})$ is of full-rank. Therefore, all diagonal elements in the diagonal matrix $\mathbf{\Lambda}_1^{1/2}\mathbf{\Lambda}_2^{-1/2}$ are non-zero.

Then, based on the uniqueness of singular value decomposition (see e.g. Proposition 4.1 in Trefethen and Bau III (1997)), $\mathbf{O}_2$ is a composition of a permutation matrix and a signature matrix. Therefore, $\mathbf{J}_{\mathbf{h}}(\mathbf{s}) = \mathbf{\Lambda}_2^{-1/2}\mathbf{O}_2\mathbf{\Sigma}(2,\mathbf{s})^{1/2}$ is a generalized permutation matrix. It thus follows that the function $\mathbf{h} := \hat{\mathbf{f}}^{-1} \circ \mathbf{f}$ is a composition of a component-wise invertible transformation and a permutation. $\square$

## A.7    Proof of Proposition 3

**Proposition 3.** *The following inequality holds*

$$n\log\left(\frac{1}{n}\sum_{i=1}^{n}\left\|\frac{\partial\hat{\mathbf{f}}^{-1}}{\partial x_i}\right\|_2\right) - \log\left|\det(\mathbf{J}_{\hat{\mathbf{f}}^{-1}}(\mathbf{x}))\right| \geq 0, \tag{8}$$

*with equality iff. $\mathbf{J}_{\hat{\mathbf{f}}^{-1}}(\mathbf{x}) = \mathbf{O}(\mathbf{x})\lambda(\mathbf{x})$, where $\mathbf{O}(\mathbf{x})$ is an orthogonal matrix and $\lambda(\mathbf{x})$ is a scalar.*

*Proof.* According to Hadamard's inequality, we have

$$\sum_{i=1}^{n}\log\left\|\frac{\partial\hat{\mathbf{f}}^{-1}}{\partial x_i}\right\|_2 - \log\left|\det(\mathbf{J}_{\hat{\mathbf{f}}^{-1}}(\mathbf{x}))\right| \geq 0, \tag{87}$$

with equality iff. vectors $\frac{\partial\hat{\mathbf{f}}^{-1}}{\partial x_i}, i = 1, 2, \ldots, n$ are orthogonal. Then applying the inequality of arithmetic and geometric means yields

$$\begin{aligned}
&n\log\left(\frac{1}{n}\sum_{i=1}^{n}\left\|\frac{\partial\hat{\mathbf{f}}^{-1}}{\partial x_i}\right\|_2\right) - \log\left|\det(\mathbf{J}_{\hat{\mathbf{f}}^{-1}}(\mathbf{x}))\right| \\
&\geq \sum_{i=1}^{n}\log\left\|\frac{\partial\hat{\mathbf{f}}^{-1}}{\partial x_i}\right\|_2 - \log\left|\det(\mathbf{J}_{\hat{\mathbf{f}}^{-1}}(\mathbf{x}))\right| \\
&\geq 0,
\end{aligned} \tag{88}$$

with the first equality iff. for all $i = 1, 2, \ldots, n$, $\left\|\frac{\partial\hat{\mathbf{f}}^{-1}}{\partial x_i}\right\|_2$ is equal. Because $\mathbf{J}_{\hat{\mathbf{f}}^{-1}}(\mathbf{x})$ is non-singular, $\frac{\partial\hat{\mathbf{f}}^{-1}}{\partial x_i}$ are orthogonal and equal to each other for all $i = 1, 2, \ldots, n$ iff. $\mathbf{J}_{\hat{\mathbf{f}}^{-1}}(\mathbf{x}) = \mathbf{O}(\mathbf{x})\lambda(\mathbf{x})$. $\square$

# B Experiments

In order to generate observational data satisfying the required assumptions, we simulate the sources and mixing process as follows:

***SS***.    To guarantee that the ground-truth nonlinear mixing process satisfies the structured sparsity condition (Assumption iv in Thm. 1), we generate observed variables with "structured" multi-layer perceptrons (MLPs): Each observed variables is only a nonlinear mixture of its own parents. For example, if the observed variable $\mathbf{x}_1$ has parents $\mathbf{s}_1$ and $\mathbf{s}_2$, then $\mathbf{x}_1 = \mathrm{MLP}(\mathbf{s}_1, \mathbf{s}_2)$. The MLP here could be replaced by any nonlinear functions.

***II***.    We generate the mixing functions based on Möbius transformations with scaled sources and constant volume. According to Liouville's results (Flanders, 1966) (also summarized in Theorem F.2 in Gresele et al. (2021)), the Möbius transformation guarantees the orthogonality between the columns vectors of its Jacobian matrix, which achieves uncorrelatedness after centering. We scaled the sources while preserving volumes before Möbius transformation with distinct scalers to make sure that the generating process is not a conformal map. We center the columns of the Jacobian as described in Appx. A.6.

***VP***.    Here we describe the generating process with a factorizable Jacobian determinant but not necessarily with orthogonal columns of Jacobian. We use a volume-preserving flow called GIN (Sorrenson et al., 2020) to generate the mixing function. GIN is a volume-preserving version of RealNVP (Dinh et al., 2016), which achieves volume preservation by setting the scaling function of the final component to the negative sum of previous ones.[5] We use the official implementation of GIN (Sorrenson et al., 2020), which is part of FrEIA.

***Base***.    Here we describe the generating process without restrictions on having a factorizable Jacobian determinant and orthogonal columns of the Jacobian. Following (Sorrenson et al., 2020), we use GLOW (Kingma and Dhariwal, 2018) to generate the mixing function. The difference between the coupling block in GLOW and GIN is that the Jacobian determinant of the former is not constrained to be one. The implementation of GLOW is also included in the official implementation of GIN (Sorrenson et al., 2020), which is also part of FrEIA.

The ground-truth sources are sampled from a multivariate Gaussian, with zero means and variances sampled from a uniform distribution on $[0.5, 3]$.[6] It is worth noting that we sample sources from a single multivariate Gaussian so that all sources are marginally independent, which is different from all previous works assuming conditional independence given auxiliary variables.

Regarding the model evaluation, we use the mean correlation coefficient (MCC) between the ground-truth and recovered latent sources. We first compute pair-wise correlation coefficients between the true sources and recovered ones. Then we solve an assignment problem to match each recovered source to the ground truth with the highest correlation between them. MCC is a standard metric to measure the degree of identifiability up to component-wise transformation in the literature (Hyvärinen and Morioka, 2016). All results are of 10 trials with different random seeds.

The sample size for the synthetic datasets is 10000. For experiments conducted on them, the learning rate is 0.01 and batch size is 1000. The number of coupling layers for both GIN and GLOW is set as 24. Regarding the image dataset, we have 25000 $32 \times 32$ images of the drawn triangle. The statistic of the dataset is described in (Yang et al., 2022), with the difference that we only use one class of triangles for unconditional priors. For experiments conducted on images, the learning rate is $3 \times 10^{-4}$ and batch size is 100. The number of coupling layers for the estimating method GIN is set as 10. The experiments are directly conducted with the official implementation of GIN (Sorrenson et al., 2020) [7] with additional regularization terms and on 4 CPU cores with 16 GB RAM.

---

[5]In the official implementation of GIN, the volume-preservation is achieved in a slightly different way compared to that in its original paper for better stability of training. There is no difference in the theoretical result w.r.t. volume-preservation.

[6]These are of the same values as previous works (Khemakhem et al., 2020; Sorrenson et al., 2020).

[7]https://github.com/VLL-HD/GIN