# OpenReview forum: "On the Identifiability of Nonlinear ICA: Sparsity and Beyond"
_NeurIPS.cc/2022/Conference — NeurIPS 2022 Accept_

### Official Review · Reviewer_3ycU · 2022-07-10

**Rating:** 8
**Confidence:** 2
**Soundness:** 3 good
**Presentation:** 4 excellent
**Contribution:** 3 good

**Summary:**

The paper proves identifiability results for Independent Component Analysis. In contrast to other recent works which exploit assumptions on auxiliary variables, this work exploits sparsity assumptions on the mixing process.

**Questions:**

See above

**Limitations:**

See above

**Strengths And Weaknesses:**

Identifiably results for ICA is an important problem as it provides a rigorous foundation for other applied areas such as disentangled feature learning and parts of causality. As such, advances in this area are important and potentially impactful.

The paper is well written and clear. It explains the results in the context of the problem and prior work. I believe it makes a solid contribution and is worth accepting to the conference.

One thing I think could improve the paper would be to elaborate more on the assumtions being made and when they are violated. In particular:
- what would happen if the ground truth data were to violate  the assumptions but the method were run anyway?
- thinking about the equivalence classes of indeterminable solutions without this assumption, is it possible to relate these equivalence classes to the sparse solutions? E.g. does each equivalence class contain exactly one such sparse solution? Do only some contain one sparse solution (and if so, what is special about these equivalence classes)?

---

> ### Author Response · Authors · 2022-08-02
> **We greatly appreciate the reviewer’s time, encouraging comments, and constructive suggestions**
>
> We greatly appreciate the reviewer’s time, encouraging comments, and constructive suggestions. Our responses are provided below.
>
> **Q1**: What would happen if the ground truth data were to violate the assumptions but the method was run anyway?
>
> **A1**: Thanks for the great question. In the ablation study, assumptions of the data generating process of *Base* (i.e., the vanilla baseline) are violated and one could observe a significant performance drop compared to others (Fig. 4). Besides, we also tried adding the regularization term during the estimation of *Base* and the performance stays low. This may not be surprising since our regularization terms are specifically designed for the proposed conditions on the generating process.
>
> **Q2**: Equivalence classes of indeterminable solutions without the assumption and the existence of the unique sparsest solution.
>
> **A2**: Thanks for the insightful comment. This is an excellent point. Theoretically, without the assumption on the mixing process, we could not identify a unique solution up to trivial indeterminacies based only on the sparsity regularization, i.e., the sparse solution may not fall within the true equivalence classes of indeterminable solutions.
>
> A straightforward example is the linear Gaussian case in which rotation indeterminacy exists. Specifically, one can easily construct different rotated solutions with equally sparse supports that lie outside of the range of acceptable indeterminacies of full identifiability. Therefore, it is possible to have non-unique sparse solutions without specific assumptions on the mixing process.
>
> Our theories provide one set of conditions for identification based on sparsity regularization. The proposed conditions could be used to formalize the pattern of equivalence classes with unique sparse solutions. At the same time, we guess that there must exist other weaker assumptions, which we hope to explore in future work because of the significance of nonlinear ICA to the foundation of unsupervised learning, disentanglement, and causal discovery.
>
> ---
>
> **Reference**:
>
> Ghassami, A., Yang, A., Kiyavash, N., & Zhang, K. (2020, November). Characterizing distribution equivalence and structure learning for cyclic and acyclic directed graphs. In International Conference on Machine Learning (pp. 3494-3504). PMLR.

---

### Official Review · Reviewer_ZHMG · 2022-07-10

**Rating:** 7
**Confidence:** 4
**Soundness:** 4 excellent
**Presentation:** 2 fair
**Contribution:** 3 good

**Summary:**

This paper considers the identifiability of the nonlinear ICA model with unconditional priors:
$x = f(s)$
where $x$ is the observed data and $s$ are the unobserved sources.

They prove that the sources $s$ are identifiable identifiable under two different settings:

1. when a "structural sparsity" assumption holds;

2. when an "independent influence" assumption holds.

They also provide experiments where they estimate the latent sources via regularized objective functions. The regularization terms are inspired by the two settings above.

More detail:

1. Structural sparsity: this assumption is that for every latent source $s_i$, there is a set of observed variables such that $s_i$ is the only latent source that participates in the generation of all variables in the set.

In Theorem 1, the authors prove that under structural sparsity, the sources can be identified up to permutation and coordinate-wise transformation.  For this result, the authors also assume invertibility of $f$, that the observed Jacobian $J_f(s)$ spans the support of $J_f(s)$, and the estimated decoder is as sparse as the true decoder.

In Theorem 3, the authors again assume structural sparsity but relax the invertibility of $f$. They prove that if the estimated $\widehat{f}$ is a 'rotated-Gaussian mixture preserving automorphism' (i.e. it has a particular functional form which rotates the true decoder $f$) then the sources are identifiable.


2. Independent influence: this assumption is that the influence of each source on the observed variables is independent of each other i.e. $\partial f / \partial s_i$ is independent of the other sources and their influences in a non-statistical sense.

In Theorem 4, the authors prove that if independent influence holds (among other assumptions) then the latent sources are identifiable.





**Questions:**

- The assumption of independent influence is that the influence of each source on the observed variables is independent of each other i.e. $\partial f / \partial s_i$ is independent of the other sources and their influences in a non-statistical sense.

Can the authors be more precise about what "in a non statistical sense" means? From the paper, it seems that it means uncorrelatedness of the Jacobian columns? Can the authors clarify?

**Limitations:**

The authors do not elaborate on real life situations where their assumptions are likely to hold (except for the animal recording example in the independent influences section).

**Strengths And Weaknesses:**

Strengths

- The paper is timely and relevant and seems like an important contribution to the study of identifiability

- The writing is clear

Weaknesses

- Some definitions are unclear (see questions below).

- The paper is missing a number of references relating to sparsity and identifiability in latent variable models.

In the linear case, Bing et al. (2020) and Rohe and Zeng (2020) prove that the assumption of sparsity in the mixing function (i.e. loadings matrix) gives identifiable solutions.

In the nonlinear case, Moran et al. (2021) prove that assuming sparsity in the decoder of a deep generative model gives identifiable solutions.

Similarly to the experiments in this paper, Rhodes and Lee (2021) propose a L1 regularization term on the Jacobian of the mixing function (without proving identifiability however).

- Bing, Xin, et al. "Adaptive estimation in structured factor models with applications to overlapping clustering." The Annals of Statistics 48.4 (2020): 2055-2081.

- Rohe, Karl, and Muzhe Zeng. "Vintage factor analysis with varimax performs statistical inference." arXiv preprint arXiv:2004.05387 (2020).

- Moran, Gemma E., et al. "Identifiable variational autoencoders via sparse decoding." arXiv preprint arXiv:2110.10804 (2021).

- Rhodes, Travers, and Daniel Lee. "Local Disentanglement in Variational Auto-Encoders Using Jacobian $ L_1 $ Regularization." Advances in Neural Information Processing Systems 34 (2021): 22708-22719.

---

> ### Author Response · Authors · 2022-08-02
> **We sincerely thank the reviewer for the time dedicated to reviewing our paper, the constructive suggestions, and encouraging feedback - Part 1**
>
> We sincerely thank the reviewer for the time dedicated to reviewing our paper, the constructive suggestions, and encouraging feedback. Please find the response to your comments and questions below.
>
> **Q1**: Several references relating to sparsity and identifiability in latent variable models are missing.
>
> **A1**: Thank you for suggesting these related references. It is extremely helpful–we have included and discussed them in the revised version. As the reviewer summarized elegantly, we would also like to note again that, though closely related, the settings of these important works differ from the nonlinear ICA setting we considered. For example, (Bing et al., 2020, Moran et al., 2021) adopt the anchor feature assumption, i.e., for each source, there are at least two observations that are influenced only by that source. This requires the number of observations to be several times larger than that of sources.
>
> **Q2**: The meaning of “in a non-statistical sense” in the assumption of independent influences.
>
> **A2**: Thanks for raising this excellent question. It means those quantities do not contain information about each other. The main reason we use ``independence in a non-statistical sense” is that statistical independence is defined w.r.t. variables, whereas the partial derivatives of the mixing function, strictly speaking, may not be considered as variables. At the same time, if we straightforwardly formalize the condition based on the uncorrelatedness of the Jacobian columns, the connection with the natural mixing might not be intuitive and lack interpretation.

---

> > ### Author Response · Authors · 2022-08-02
> > **We sincerely thank the reviewer for the time dedicated to reviewing our paper, the constructive suggestions, and encouraging feedback - Part 2**
> >
> > **Others**: Real-life situations for the sparsity assumption.
> >
> > **A3**: We are very grateful for your constructive feedback. We have added a section of additional discussion on it (Appx. C in the appendix) as follows:
> >
> > ---
> >
> > ``Sparsity assumptions have been widely used in various fields. For latent variable models, sparsity in the generating process plays an important role in the disentanglement or identification of latent factors both empirically and theoretically (Bing et al., 2020; Rohe and Zeng, 2020; Moran et al., 2021; Rhodes and Lee, 2021; Lachapelle et al., 2022). In causality, various versions of Occam's razor, such as faithfulness (Spirtes et al., 2000), the minimality principle (Zhang, 2013), and frugality (Raskutti and Uhler, 2018; Forster et al., 2020), have been proposed to serve as fundamental assumptions for identifying the underlying causal structure.
> >
> > Formulated as a measure of the density of dependencies, sparsity assumptions are more likely to be held when the observations are actually influenced by the sources in a ``simple” way. For example, in biology, analyses on ecological, gene-regulatory, metabolic, and other living systems find that active interactions may often be rather sparse (Busiello et al., 2017), even when these systems evolve with an unlimited number of complicated external stimuli. In physics, it is an important heuristic that a relatively small set of laws govern complicated observed phenomena. For instance, Einstein's theory of special relativity contains parsimonious relations between substances as an important heuristic to shave away the influence of ether compared to Lorentz's theory (Einstein, 1905; Nash, 1963).
> >
> > However, sparsity is not an irrefutable principle and our assumption may fail in a number of situations. The most direct one could be a scenario with heavily entangled relations between sources and observations. Let us consider the example of animal filmmaking in Sec. 4, where people are recording the sound of animals in a safari park. If the filming location is restricted to a narrow area of the safari park and multiple microphones are gathered together, the recording of each microphone will likely be influenced by almost all animals. In such a case, the dependencies between the recording of microphones and the animals are rather dense and our sparsity assumption is most likely not valid.
> >
> > At the same time, even when the principle of simplicity holds, our formulation of sparsity, which is based on the sparse interactions between sources and observations, may still fail. One reason for this is the disparity between mechanism simplicity and structural sparsity. To illustrate this, one could consider the effect of sunlight on the shadow angles at the same location. In this case, the sun’s rays and the shadow angles act as the sources and the observations, respectively. Because rays of sunlight, loosely speaking, may be parallel to each other, the processes of them influencing the shadow angles may be almost identical. Thus, the influencing mechanism could be rather simple. On the other hand, each shadow angle is influenced by an unlimited number of the sun’s rays, which indicates that the interactions between them may not be sparse, therefore violating our assumption. This sheds light on one of the limitations of our sparsity assumption, because the principle of simplicity could be formulated in several ways. Besides, these different formulations also suggest various possibilities for identifiability based on simplicity assumptions. Another proposed assumption, i.e., independent influences, may be one of the alternative formulations, and more works remain to be explored in the future.”
> >
> > ---

---

> > > ### Author Response · Authors · 2022-08-02
> > > **We sincerely thank the reviewer for the time dedicated to reviewing our paper, the constructive suggestions, and encouraging feedback - Part 3**
> > >
> > > **References**:
> > >
> > > Bing, X., Bunea, F., Ning, Y., & Wegkamp, M. (2020). Adaptive estimation in structured factor models with applications to overlapping clustering. The Annals of Statistics, 48(4), 2055-2081.
> > >
> > > Moran, G. E., Sridhar, D., Wang, Y., & Blei, D. M. (2021). Identifiable variational autoencoders via sparse decoding. arXiv preprint arXiv:2110.10804.
> > >
> > > K. Rohe and M. Zeng. Vintage factor analysis with varimax performs statistical inference. arXiv preprint arXiv:2004.05387, 2020.
> > >
> > > T. Rhodes and D. Lee. Local disentanglement in variational auto-encoders using jacobian l_1 regularization. Advances in Neural Information Processing Systems, 34:22708–22719, 2021.
> > >
> > > S. Lachapelle, P. R. López, Y. Sharma, K. Everett, R. L. Priol, A. Lacoste, and S. Lacoste-Julien. Disentanglement via mechanism sparsity regularization: A new principle for nonlinear ICA. Conference on Causal Learning and Reasoning, 2022.
> > >
> > > P. Spirtes, C. N. Glymour, R. Scheines, and D. Heckerman. Causation, prediction, and search. MIT press, 2000.
> > >
> > > J. Zhang. A comparison of three occam’s razors for markovian causal models. The British journal for the philosophy of science, 64(2):423–448, 2013.
> > >
> > > G. Raskutti and C. Uhler. Learning directed acyclic graph models based on sparsest permutations. Stat, 7(1):e183, 2018.
> > >
> > > M. Forster, G. Raskutti, R. Stern, and N. Weinberger. The frugal inference of causal relations. The British Journal for the Philosophy of Science, 2020.
> > >
> > > D. M. Busiello, S. Suweis, J. Hidalgo, and A. Maritan. Explorability and the origin of network sparsity in living systems. Scientific reports, 7(1):1–8, 2017.
> > >
> > > A. Einstein. Does the inertia of a body depend upon its energy-content. Annalen der Physik, 18(13): 639–641, 1905.
> > >
> > > L. K. Nash. The nature of the natural sciences. 1963.

---

### Official Review · Reviewer_TtGx · 2022-07-11

**Rating:** 7
**Confidence:** 3
**Soundness:** 3 good
**Presentation:** 2 fair
**Contribution:** 3 good

**Summary:**

This paper presents new identifiability result for nonlinear ICA that gets around the need for auxiliary information by leveraging a sparsity assumption on the mixing function. While sparsity has been explored in the nonlinear ICA community, the existing results have focused on the relationship between source distributions (Lachapelle et al., 2022), rather than on sparsity of the mixing function. On exception is Gresele et al [2021] that did show a different sparsity assumption rules out some of the standard counterexamples, but didn't provide a full identification proof. This paper does that and shows analogous results for the linear Gaussian case which well-known to not be identifiable without additional assumptions (usually non-Gaussian sources). The paper also gives results under an "independent influences" assumption which is more similar to Gresele et al [2021] and show identification results there too. Finally, the theory results are supported with some simple experiments.

**Questions:**

1. Do you know what was preventing MCC scores from getting above 0.9? These problems are relatively straightforward so I would have expected to see MCC score $\rightarrow 1$; it would be good to understand what the limitations of the empirical results are?

Minor
* Condition (iii) of theorem 1 is hard to parse without a lot of back references. Consider rewording as follows, "Denote by F the support of (all possible values of **the Jacobian of $f$**) $J_f (s)$. For all $i \in {1, . . . , n}$,. there exist **|F_{i,:} | sources**...". Similar amendments could be made to the other theorem statements too - the conditions in the theorems are already pretty dense, so it helps to remind the reader what all the variables refer to.

**Limitations:**

Yes.

**Strengths And Weaknesses:**

Strengths
* Strong identification (up to a component-wise transform & permutation) results in a hard problem.

Weakness
* I am always very uncomfortable with making assumptions on the mixing function $f: s \rightarrow x$, especially in the image domain. While it is relatively intuitive that there should be some simple dependencies among the source variables (as people, we wouldn't be able to understand the world if this were not the case), it's not obvious that the mixing function should have any sparsity: consider how realistic images are rendered in modern rendering engines. The most realistic looking renderers use ray-tracing which explicitly bounce light around all the objects in the scene in order to correctly model how light reflects in a scene. That seems to imply a dense dependence between sources rather than any kind of sparsity. Of course, it may be the case that structured sparsity is a reasonable approximation of the mixing function, but I would have liked to see some more discussion on when we can expect this to hold. The discussion on page 7 with reference to the independent influences assumption was far better.
* The empirical results have the expected relationships between approaches, but I was surprised that the MCC scores were only $\approx 0.8$.

I am conflicted on this paper - while ICA is not my research area, I know that this is a very well-studied area and identification in this IID setting is hard to achieve (as the paper points out - most papers rely on auxiliary variables). So if we take this work as a pure piece of theory, the structured sparsity condition is a sufficient condition for identification is a difficult domain, and this is very interesting paper. On the other hand - the practitioner in me can barely parse the structured sparsity assumption - so I'd have no idea whether or not it applied in my domain, so I think a lot more work is needed in expanding on when we expect this assumption to hold or alternatively, are there any testable implications of this assumption?

---

> ### Author Response · Authors · 2022-08-02
> **We are very grateful for your time, insightful comments, and encouragement - Part 1**
>
> We are very grateful for your time, insightful comments, and encouragement. Below please see our point-by-point response.
>
> **Q1**: More discussion of the assumption of sparsity on the mixing function.
>
> **A1**: We sincerely appreciate this essential point. We completely agree with you that the sparsity assumption might not hold in some types of situations, such as the ray tracing-based rendering you mentioned. Regarding the testability of the assumption, we agree that it is rather challenging to determine whether the assumption holds.  For instance, assumptions such as faithfulness (Spirtes et al., 2000) and frugality (Raskutti and Uhler, 2018; Forster et al., 2020) are generally not testable (specifically, frugality can be interpreted as sparsity of the edges in the causal structure), since the true structures are not known a priori, which therefore remains an active area of research. It is worth noting that some parts of the faithfulness assumption are shown to be partially testable–Ramsey et al. (2006) showed that the orientation faithfulness assumption is testable under the adjacency faithfulness assumption. Therefore, we think it is an interesting future direction to explore whether this type of partial testability can be carried out for the sparsity assumption in our work.
>
> Regarding the practical significance and limitation of the sparsity assumption, we added a separate section in Appendix (Appx. C) in light of your suggestions. The discussion is as follows:
>
> ---
> ``Sparsity assumptions have been widely used in various fields. For latent variable models, sparsity in the generating process plays an important role in the disentanglement or identification of latent factors both empirically and theoretically (Bing et al., 2020; Rohe and Zeng, 2020; Moran et al., 2021; Rhodes and Lee, 2021; Lachapelle et al., 2022). In causality, various versions of Occam's razor, such as faithfulness (Spirtes et al., 2000), the minimality principle (Zhang, 2013), and frugality (Raskutti and Uhler, 2018; Forster et al., 2020), have been proposed to serve as fundamental assumptions for identifying the underlying causal structure.
>
> Formulated as a measure of the density of dependencies, sparsity assumptions are more likely to be held when the observations are actually influenced by the sources in a ``simple” way. For example, in biology, analyses on ecological, gene-regulatory, metabolic, and other living systems find that active interactions may often be rather sparse (Busiello et al., 2017), even when these systems evolve with an unlimited number of complicated external stimuli. In physics, it is an important heuristic that a relatively small set of laws govern complicated observed phenomena. For instance, Einstein's theory of special relativity contains parsimonious relations between substances as an important heuristic to shave away the influence of ether compared to Lorentz's theory (Einstein, 1905; Nash, 1963).
>
> However, sparsity is not an irrefutable principle and our assumption may fail in a number of situations. The most direct one could be a scenario with heavily entangled relations between sources and observations. Let us consider the example of animal filmmaking in Sec. 4, where people are recording the sound of animals in a safari park. If the filming location is restricted to a narrow area of the safari park and multiple microphones are gathered together, the recording of each microphone will likely be influenced by almost all animals. In such a case, the dependencies between the recording of microphones and the animals are rather dense and our sparsity assumption is most likely not valid.
>
> At the same time, even when the principle of simplicity holds, our formulation of sparsity, which is based on the sparse interactions between sources and observations, may still fail. One reason for this is the disparity between mechanism simplicity and structural sparsity. To illustrate this, one could consider the effect of sunlight on the shadow angles at the same location. In this case, the sun’s rays and the shadow angles act as the sources and the observations, respectively. Because rays of sunlight, loosely speaking, may be parallel to each other, the processes of them influencing the shadow angles may be almost identical. Thus, the influencing mechanism could be rather simple. On the other hand, each shadow angle is influenced by an unlimited number of the sun’s rays, which indicates that the interactions between them may not be sparse, therefore violating our assumption. This sheds light on one of the limitations of our sparsity assumption, because the principle of simplicity could be formulated in several ways. Besides, these different formulations also suggest various possibilities for identifiability based on simplicity assumptions. Another proposed assumption, i.e., independent influences, may be one of the alternative formulations, and more works remain to be explored in the future.”
>
> ---

---

> > ### Author Response · Authors · 2022-08-02
> > **We are very grateful for your time, insightful comments, and encouragement - Part 2**
> >
> > **Q2**: What was preventing MCC score $\rightarrow$ 1?
> >
> > **A2**: Thanks for the great question. During the estimation, we used General Incompressible flow Network (GIN) to maximize the objective function, of which the coupling block consists of MLPs and ReLU activation layers. Thus, the estimation process may never achieve the global optimum within finite epochs as the overall optimization problem is nonconvex, as is typical for methods involving deep learning. Besides, the error due to the finite sample size may also prevent the model from reaching a perfect performance. Moreover, to induce sparsity, we use $L_1$ penalty as an approximation of the $L_0$ penalty to enable gradient-based optimization, which may also introduce errors empirically. Based on our empirical observations as well as related works focusing on these issues, we believe that these are part of the underlying reasons.
> >
> > **Minor**: Polishing condition (iii) of theorem 1 to avoid potential back-references by the readers.
> >
> > **A3**: Thanks for your constructive suggestion. We have updated it as well as some other conditions in the paper.
> >
> > ---
> >
> > **References**:
> >
> > P. Spirtes, C. N. Glymour, R. Scheines, and D. Heckerman. Causation, prediction, and search. MIT press, 2000.
> >
> > G. Raskutti and C. Uhler. Learning directed acyclic graph models based on sparsest permutations. Stat, 7(1):e183, 2018.
> >
> > M. Forster, G. Raskutti, R. Stern, and N. Weinberger. The frugal inference of causal relations. The British Journal for the Philosophy of Science, 2020.
> >
> > Ramsey, J., Zhang, J., & Spirtes, P. L. (2012). Adjacency-faithfulness and conservative causal inference. arXiv preprint arXiv:1206.6843.
> >
> > Bing, X., Bunea, F., Ning, Y., & Wegkamp, M. (2020). Adaptive estimation in structured factor models with applications to overlapping clustering. The Annals of Statistics, 48(4), 2055-2081.
> >
> > K. Rohe and M. Zeng. Vintage factor analysis with varimax performs statistical inference. arXiv preprint arXiv:2004.05387, 2020.
> >
> > Moran, G. E., Sridhar, D., Wang, Y., & Blei, D. M. (2021). Identifiable variational autoencoders via sparse decoding. arXiv preprint arXiv:2110.10804.
> >
> > T. Rhodes and D. Lee. Local disentanglement in variational auto-encoders using jacobian l_1 regularization. Advances in Neural Information Processing Systems, 34:22708–22719, 2021.
> >
> > S. Lachapelle, P. R. López, Y. Sharma, K. Everett, R. L. Priol, A. Lacoste, and S. Lacoste-Julien. Disentanglement via mechanism sparsity regularization: A new principle for nonlinear ICA. Conference on Causal Learning and Reasoning, 2022.
> >
> > J. Zhang. A comparison of three occam’s razors for markovian causal models. The British journal for the philosophy of science, 64(2):423–448, 2013.
> >
> > D. M. Busiello, S. Suweis, J. Hidalgo, and A. Maritan. Explorability and the origin of network sparsity in living systems. Scientific reports, 7(1):1–8, 2017.
> >
> > A. Einstein. Does the inertia of a body depend upon its energy-content. Annalen der Physik, 18(13): 639–641, 1905.
> >
> > L. K. Nash. The nature of the natural sciences. 1963.

---

### Meta-Review · Area_Chair_6nAk · 2022-08-26

**Recommendation:** Accept
**Confidence:** Certain

**Metareview:**

Strong paper with all reviewers arguing for acceptance.

Only minor concerns from the reviewers were on whether the preconditions required for the theorems in the paper were likely to hold in practice. This was discussed thoughtfully by the author response, including a new appendix section.

**Award:**

No

---

### Decision · Program_Chairs · 2022-09-14

Accept